



# Sources of nitrous oxide and fate of mineral nitrogen in sub-Arctic permafrost peat soils

Jenie A. Gil[1,2], Maija E. Marushchak[3,] Tobias Rütting[4], Elizabeth M. Baggs[5], Tibisay Pérez[6], Alexander Novakovskiy[7], Tatiana Trubnikova[1], Dmitry Kaverin[7], Pertti J. Martikainen[1] and Christina Biasi[1]

[1]Department of Environmental and Biological Sciences, University of Eastern Finland, Kuopio, P.O. Box 1627, FI-70211, Finland.
[2]Department of Integrative Biology, Great Lakes Bioenergy Research Center, Michigan State University, 288 Natural Science Bldg., East Lansing, MI 48824-1302, USA.
[3]Department of Biological and Environmental Science, University of Jyväskylä, P.O. Box 35, FI-40014 University of
Jyväskylä, Finland.
[4]Department of Earth Sciences, University of Gothenburg, Box 460, 405 30 Gothenburg, Sweden.
[5]Global Academy of Agriculture and Food Security, The Royal (Dick) School of Veterinary Studies, University of Edinburgh, Easter Bush Campus, Midlothian EH25 9RG, UK.
[6]Centro de Ciencias Atmosféricas y Biogeoquímica. IVIC. Aptdo. 20634. Caracas 1020A. Venezuela.
[7]Institute of Biology, Komi SC UB RAS, 167982 Syktyvkar, Russia.

*Correspondence to:* J. Gil (gillugoj@msu.edu)

**Abstract**

Nitrous oxide ($N_2O$) emissions from permafrost-affected terrestrial ecosystems have received little attention, largely because they have been thought to be negligible. Recent studies, however, have shown that there are habitats in subarctic tundra
emitting $N_2O$ at high rates, such as bare peat surfaces on permafrost peatlands. The processes behind $N_2O$ production in these high-emitting habitats are, however, poorly understood. In this study, we established an *in situ* [15]N-labelling experiment with the main objectives to partition the microbial sources of $N_2O$ emitted from bare peat surfaces (BP) on permafrost peatlands and to study the fate of ammonium and nitrate in these soils and in adjacent vegetated peat surfaces (VP) showing low $N_2O$ emissions. Our results confirm the hypothesis that denitrification is mostly responsible for the high $N_2O$ emissions from BP
surfaces. During the study period denitrification contributed with ~79% of the total $N_2O$ emission in BP, while the contribution of ammonia oxidation was less, about 19 %. However, nitrification is a key process for the overall $N_2O$ production in these soils with negligible external nitrogen (N) load because it is responsible for nitrite/nitrate supply for denitrification, as also supported by relatively high gross nitrification rates in BP. Generally, both gross N mineralization and gross nitrification rates were much higher in BP with high $N_2O$ emissions than in VP, where the high C/N ratio together with low water content was
likely limiting N mineralization and nitrification and, consequently, $N_2O$ production. Also, competition for mineral N between plants and microbes was additionally limiting N availability for $N_2O$ production in VP. Our results show that multiple factors control $N_2O$ production in permafrost peatlands, the absence of plants being a key factor together with inter-mediate to high water content and low C/N ratio, all factors which also impact on gross N turnover rates. The intermediate to high soil water content which creates anaerobic microsites in BP is a key $N_2O$ emission driver for the prevalence of denitrification to occur.
This knowledge is important for evaluating future permafrost –N feedback loops from the Arctic.

Keywords: Permafrost soils, Arctic, sub-Arctic, soils, nitrous oxide emissions, [15]N-labelling experiment, source partitioning, denitrification, nitrification, gross N turnover rates, permafrost-climate feedbacks



## 1 Introduction

The Arctic and sub-Arctic regions store more than 50% of the Earth's soil carbon (C) pool (1330–1580 Pg) (Schuur *et al.,* 2015). The possible increase in release of the greenhouse gases carbon dioxide ($CO_2$) and methane ($CH_4$) from these carbon stocks to the atmosphere under a changing climate has been intensively studied (Schuur *et al.,* 2009; Schuur *et al.,* 2015;

Schädel *et al.,* 2016). However, Arctic soils store not only a huge amount of C but has also a large nitrogen (N) reservoir (conservative estimate for 0-3 m: 67 Pg N) (Harden *et al.,* 2012), but little is known of the potential of this N to be released as the strong greenhouse gas nitrous oxide ($N_2O$). Although soils world-wide are important $N_2O$ sources responsible for 60% of the global emissions (IPCC, 2013), it has traditionally been suggested that $N_2O$ emissions from Arctic soils are negligible because of their low concentrations of mineral N (Ma *et al., 2007*; Takakai *et al.,* 2008; Siciliano *et al.,* 2009; Goldberg *et al.,*

2010). However, this generalization has been challenged by the identification of hotspots of $N_2O$ on raised permafrost peatlands (Repo *et al.,* 2009; Marushchak *et al.*, 2011) and by measurements of high $N_2O$ concentrations in upland tundra soils facing thermokarst formation following permafrost thaw (Abbott & Jones, 2015). A field warming experiment in an permafrost peatland further showed that soil warming (average increase of 0.95°C) promotes $N_2O$ release not just from bare peat hotspots, but also from adjacent vegetated surfaces that do not emit $N_2O$ under the present climate (Voigt *et al.,* 2017a).

In addition, results from mesocosms and soil incubation studies show that arctic soils have potential for high $N_2O$ emissions after permafrost thawing (~ 3 - 4 mg $N_2O$. $m^{-2}$ $d^{-1}$, Elberling *et al.,* 2010; Voigt *et al*., 2017b). In a recent review it was concluded that  the emissions of $N_2O$ from permafrost soils could be up to 1.27 Tg $N_2O$-N $yr^{-1}$, which represents 11.6 % of total $N_2O$ emissions from natural soils (Voigt *et al.,* 2020). Thus, $N_2O$ emissions from permafrost soils cannot be ignored anymore.

Even though there is increasing evidence of $N_2O$ production from permafrost soils, with potential global importance (Voigt *et al.,* 2020), mechanisms underlying the release of this strong greenhouse gas remain largely unclear. A better understanding of $N_2O$ production from permafrost soils is needed to evaluate the role the Arctic and sub-Arctic play in the global $N_2O$ budget at present and in future. Under the present climate, $N_2O$ emissions from bare surfaces of permafrost peatland (-0.24 to 31 mg $N_2O$ $m^{-2}$ $d^{-1}$) (Repo *et al.,* 2009; Voigt *et al.,* 2017a; Gil *et al.,* 2017) – the until now largest known sources of $N_2O$ from the

Arctic - can achieve similar magnitudes as those from temperate and boreal agricultural soils (Maljanen *et al.,* 2010) and tropical forests soils (Werner *et al.,* 2004). It is thought that these hotspots have developed through frost action and wind erosion (Kaverin *et al.* 2013). The absence of vegetation together with low C/N ratio and intermediate soil water content (~60% water-filled pore space; WFPS) have been suggested to be the key environmental factors associated with the high $N_2O$ emissions from these bare peat surfaces (Repo *et al*. 2009). Generally, the main processes responsible for $N_2O$ production in

soils are nitrification (ammonia oxidation) and the nitrate reducing pathway of denitrification which tend to predominate under suboxic and anaerobic conditions, respectively (Baggs, 2011). In unfertilized, natural ecosystems with low atmospheric deposition of N like the Arctic, nitrate produced during nitrification is the main N source for denitrification. Therefore, these two processes are tightly coupled in Arctic soil (Siljanen *et al*. 2019). Low C/N ratios of the bulk soil in these systems (23 +/- 2; Repo et al., 2009) may favor net N mineralization and nitrification, and intermediate soil water status may allow both

aerobic (including nitrification) and anaerobic (including denitrification) processes to take place simultaneously. The lack of vegetation and consequently N uptake by plants means better availability of mineral N for soil microbes. All in all, the bare peat environment can be considered conducive to microbial $N_2O$ production both in nitrification and denitrification.



Although we have some understanding on the factors controlling N turnover and N availability for microbes in the permafrost peat soils, the role of various microbial processes in $N_2O$ production in these soils is still limited. To get more information on these processes is important to be able to predict responses of $N_2O$ emissions from Arctic ecosystems to climate induced changes. For example, increase in soil water content as it is predicted e.g., for Alaska and likely also elsewhere in the Arctic

(Douglas *et al.,* 2020) potentially affects the microbial processes. Nitrification and denitrification are controlled by different environmental factors. Denitrification releases usually more $N_2O$ under wetter, more anaerobic conditions and has been suggested as the key process for $N_2O$ production in bare peat surfaces (Repo *et al.,* 2009). This is supported by results from laboratory incubations where nitrate addition stimulated $N_2O$ production under anoxic conditions (Palmer *et al.,* 2011). However, isotope analysis of $N_2O$ ([15]N natural abundance, site preference values) from these hotspots in tundra in a dry year.

with low net emissions suggested that ammonia oxidizing nitrifiers could play a major role in dry conditions (Gil *et al.,* 2017). However, the limitations of such natural abundance approaches are well documented (Decock and Six, 2013; Toyoda *et al.,* 2015; Gil *et al.,* 2017), and include overlapping source signals and changing isotope fingerprints under variable environmental conditions. To overcome them, [15]N-enrichment approaches provide the ability to quantify and distinguish microbial sources of $N_2O$ *in situ*, particularly ammonia oxidation and denitrification (Stevens *et al.*, 1997; Baggs, 2011). This approach also

enables tracing of [15]N through the plant-soil system, providing valuable information on N processes including gross turnover rates and N uptake into plants (Gardner *et al.,* 2009; Harrison *et al.,* 2012; Wild *et al.,* 2015). Particularly data on gross N turnover rates including gross ammonification and nitrification are rare from the Arctic.

In this study, we conducted an *in situ* [15]N-enrichment experiment using a virtual core injection technique (Rütting *et al.,* 2011). Our objectives were, first, to partition between denitrification and nitrification as sources of $N_2O$ emitted from the $N_2O$

hotspots (bare peat; BP) located on permafrost peatlands, and second, to trace the fate of applied [15]N in BP and adjacent vegetated peat (VP). VP has shown low $N_2O$ emissions in previous studies. We hypothesized that (1) denitrification is the predominant pathway of $N_2O$ production in the BP, if emissions are high under typical climatic conditions, (2) a major proportion of the added [15]N is released as nitrogenous gases from BP but in VP immobilization is the most important sink of N, indicating that competition for N is a key regulator of $N_2O$ in these peatlands, and that (3) in addition to the absence of

vegetation, lower C/N ratios and higher water content support higher N turnover rates in BP (as compared to VP), and are important factors leading to higher $N_2O$ fluxes there.

## 2 Materials and methods

### 2.1 Study site and soil characteristics

The experiment was carried out at the Seida study site which is located in sub-Arctic northwestern Russia (67º03'N, 62º57'E)

in the discontinuous permafrost zone. The landscape in the area is fragmented and consists of many ecosystem types including tundra heath, peat plateaus (raised, ombrotrophic peatlands uplifted by ice-rich permafrost) and water-logged fens and bogs. The bare peat surfaces, located on the peat plateau, are round in shape with an average diameter of 20 m, and have only sporadic bryophytes and lichens. The growing season in the study region lasts approximately 3 months, from mid-late June to early-mid September. The mean annual precipitation is 505 mm and the mean annual air temperature is -5.8 °C. The

warmest month is usually July with a mean air temperature of 12.5 °C followed by August with 9.4 °C (30 years averages,





data from weather station at Vorkuta (67º48'N, 64º01'E); Komi Republican Center for Hydrometeorological and Environmental Monitoring). The mean precipitation sum for the period July – September is 121 mm. Additional information on the site characteristics and climatic conditions can be found in Repo *et al.,* (2009); Marushchak *et al.* (2013) and Biasi *et al.,* (2014). In 2010, when our study was undertaken, the warmest month was July with a mean air temperature of 12.9 °C,

similar to the long-term mean, while August was warmer than the long-term mean. The maximum daily air temperature (22 °C) was registered in August. The cumulative precipitation for the period July - September was close to the long-term average, 113 mm. Most of the rainfall took place during mid- August, which resulted in high soil water content at this time.

BP surfaces consist mainly of decomposed fen peat. VP surfaces have typical bog vegetation including vascular plants such as *Ledum decumbes, rubus chamaeomorus, Vaccinium uliginosum*, and mosses (e.g., *Sphagnum, Dicranum sp.)* and lichens

(e.g., *Cladina sp.*) (Table 1).

**Table 1**. Soil characteristics of the topsoil (0-10cm) of the bare peat (BP) and vegetated peat (VP) surfaces

| Soil type | pH | BD (g cm⁻³) | SOM (%) | %C | %N | C/N | [NO₃⁻] (mg N Kg⁻¹ DW) | [NH₄⁺] (mg N Kg⁻¹ DW) | WFPS (%) | Max. seasonal thaw depth (cm) |
|---|---|---|---|---|---|---|---|---|---|---|
| **BP** | $3.2 \pm 0.1$ | $0.27 \pm 0.02^a$ | $96 \pm 2^a$ | $54 \pm 6^a$ | $2.2 \pm 0.3^a$ | $23 \pm 2^a$ | $60 \pm 11^a$ | $116 \pm 39^a$ | $67 \pm 5^a$ | $70 \pm 5^a$ |
| **VP** | $3.4 \pm 0.1$ | $0.05 \pm 0.02^b$ | $98 \pm 1^b$ | $47 \pm 2^b$ | $0.8 \pm 0.2^b$ | $62 \pm 16^b$ | $11 \pm 4^b$ | $35 \pm 6^b$ | $30 \pm 7^b$ | $60 \pm 12^b$ |

Values are mean ± 1 SE, for the sampling period during the growing season 2007,2008 (Marushchak *et al*., 2011) and 2010 (this study). (n=3 for each soil type).

Different letters indicate statistically significant differences between surface types (p < 0.005).

## 2.2 ¹⁵N-enrichment experiment

### 2.2.1 Experimental design

The study took place during the growing season 2010, starting on July 21 and ending on August 13 (24 days). The ¹⁵N labelling

experiment was conducted *in situ* on BP and adjacent VP in three replicates per treatment type (n=3). The soil surfaces were selected based on their contrasting N₂O emission rates reported in previous field campaigns at the site (2007-2008; Repo *et al.,* 2009; Marushchak *et al.,* 2011). Bare peat surfaces are known to act as N₂O hotspots in contrast to VP where the N₂O fluxes are low.

The experiment comprised three different ¹⁵N-labelling treatments with either single or double ¹⁵N-labelling: ¹⁴NH₄¹⁵NO₃

(Treatment 1; T1), ¹⁵NH₄¹⁴NO₃ (Treatment 2; T2) and ¹⁵NH₄¹⁵NO₃ (Treatment 3; T3) with each of the labelled moieties being applied at 98 atom % ¹⁵N. We used T1 with ¹⁵N-NO₃⁻ label to calculate gross nitrification and to quantify N₂O emissions produced by nitrate reduction, which we attribute here to denitrification, while N₂O from nitrification was estimated using the difference in ¹⁵N-N₂O between the T3 (¹⁵NH₄¹⁵NO₃) and T1 (¹⁴NH₄¹⁵NO₃). Treatment 2 (¹⁵NH₄¹⁴NO₃) was added to calculate



gross mineralization and to account for $^{15}$N-N$_2$O fluxes from $^{15}$N-NO$_3^-$ which was nitrified after addition of $^{15}$N-NH$_4^+$ (Baggs *et al.,* 2003; see more details below). The calculations are based on the assumptions of negligible nitrate ammonification and re-mineralization of $^{15}$NH$_4$ within the first 72 hours (Braun *et al.*, 2018). The application rate of the label solutions was based on soil inorganic N concentrations in 2007-2008. For VP surfaces the label solutions were applied at a rate of 5 mg NO$_3^-$-N Kg$^{-1}$ dry soil (1 µg N cm$^{-2}$) and 17 mg NH$_4^+$-N Kg$^{-1}$ dry soil (2 µg N cm$^{-2}$) while for BP surfaces the application rates were 30 mg NO$_3^-$-N Kg$^{-1}$ dry soil (10 µg N cm$^{-2}$) and 58 NH$_4^+$-N Kg$^{-1}$ dry soil (20 µg N cm$^{-2}$). These rates are equivalent to approximately 50% of the native soil extractable N pools in the soils during the growing season (Table 1). The total quantity of mineral N added never exceeded maximum NO$_3^-$ or NH$_4^+$ concentrations found in the native, unamended soils. All treatments were applied to a depth at 6 cm and the application rates were calculated based on known bulk densities (Table 1).

The $^{15}$N solutions were added *in situ* adopting the virtual core injection technique of Rütting *et al.,* (2011). For the $^{15}$N labelling applications and samplings a 20 cm × 20 cm sub-plot was demarcated in each plot. Inside these sub-plots, a smaller area (16 cm x16 cm) was marked and a template was used for N addition and soil sampling. For the injection of $^{15}$N solutions, 49 syringes (1 mL) were used with the template to release the $^{15}$N solutions from the syringes into the soil as uniformly as possible (both horizontally and vertically). Since the $^{15}$N-labelled areas were to be sampled destructively for each time/sampling point, the injections were repeated 7 times in each replicate rotating the template around the plot. We thus carried out a total of 126 injections (2 surface type (BP and VP) x 3 replicates per surface type x 3 $^{15}$N treatments (NH$_4$$^{15}$NO$_3$, $^{15}$NH$_4$NO$_3$ and $^{15}$NH$_4$$^{15}$NO3) x 7 sampling time points). For logistical reasons, the $^{15}$N applications were completed in two days (from 21 and 22 July 2010) but both soil surface types (VP and BP) were always labelled at the same time.

After the $^{15}$N addition, the following samples were taken at 0, 1 h, 24 h and 3, 5, 9, 15, 24 days: Surface gas flux samples for N$_2$O (concentration) and $^{15}$N-N$_2$O (isotopic) analyses and soil samples for mineral N (NH$_4^+$ and NO$_3^-$), total N (TN) and their $^{15}$N-enrichments. In addition, all above ground plants as well as roots were collected from VP surfaces on the same days. All samples were analyzed for N concentrations and $^{15}$N-emrichments as described below.

### 2.2.2 Gas sampling and analysis

Emissions of $^{15}$N-N$_2$O were determined using static chambers (Heikkinen *et al*, 2002). A collar was inserted in the soil one hour before the gas sampling and a small PVC chamber (diameter 10 cm, volume 920 cm$^3$) was attached when collecting the gases. The chamber had an inlet (polyamide nylon tube) equipped with a three-way stopcock (Steritex ®3W) for gas sampling. A gas sample (ambient, t = 0) was taken near the collar before closing the chamber and 40 minutes after the chamber was placed on the collar, using a 100 mL polypropylene syringe with Luer lock tip (Terumo®, Tokyo, Japan) fitted with a three-way stopcock (Steritex ®3W, CODAN Limited, UK). Temperature inside the chamber was recorded at the beginning and at the end of each closure period. Gas samples were transferred into 12 ml exetainers equipped with butyl rubber septa (Labco Ltd, UK) the same day of sampling. Concentrations of N$_2$O were analyzed 1-2 months later at the University of Eastern Finland. A leakage test with a standard gas showed that leakage for N$_2$O was negligible ($\leq$ 3% over the storage period). The concentration of N$_2$O was measured with a gas chromatograph as described in Gil *et. al,* (2017). The N$_2$O fluxes were calculated from the difference in concentration at the start (t = 0) and end of closure (t = 40 min). This methodology had first been tested and validated for linearity of gas accumulation over the 40 min timescale. Emission rates obtained from these small chambers with one endpoint sampling and non-permanent collars were compared with emissions measured by the static





chamber technique with 4-5 sampling points in 40 min used at the site during this experiment period and previous sampling campaigns (Fig. 1) (Repo *et al.,* 2009; Marushchak *et al.,* 2011, Gil *et al.,* 2017).

Samples for $^{15}$N-N$_2$O determination were taken at the end of the enclosure of 40 min and stored in 60 ml gas-tight glass flasks (Supelco, UK) and their $^{14}$N/$^{15}$N ratios determined at the Stable Isotope Facility at the University of California, Davis, using

a Delta V Plus isotope ratio mass spectrometer (IRMS) operated in continuous flow mode (Thermo Scientific, Bremen, Germany) coupled with an online pre-concentrator and a GasBench (Thermo Finnigan, Bremen, Germany). The $^{15}$N-N$_2$O flux rates were calculated from the atom % excess $^{15}$N of the samples and linear regression slopes (at%$^{15}$N-N$_2$O excess vs. time).

### 2.2.3 Soil sampling and analyses

Immediately after gas sampling, soil samples were taken by pushing a PVC tube (length: 10 cm; diameter: 5 cm; volume: 70

cm$^3$) into the soil (0 – 6 cm) at the center of the labeling subplot area. Soil samples were sieved, homogenized, and extracted with KCl (2$M$) on the day of collection and extracts were preserved frozen for later analysis of concentrations of NH$_4^+$-N and NO$_3^-$-N and their $^{15}$N-enrichments. A subsample of the soil was dried at 60 °C and preserved for later analysis of total N and its $^{15}$N-enrichment. The concentrations of NH$_4^+$ and NO$_3^-$ in the extracts were measured by spectrophotometry (Wallac-Data Analyzer) using microtiter plate format and following the protocol of Fawcett and Scott (1960) for NH$_4^+$ (630 nm) and Griess

method for NO$_3^-$ (544 nm) (Miranda *et al.,* 2001). The $^{15}$N-enrichment in mineral N was determined by the micro-diffusion method (Herman *et al.,* 1995) and analyzed on an elemental analyzer coupled to an isotope ratio mass spectrometer (EA-IRMS), which included a Thermo Finnigan DELTA XP Plus IRMS, Flash EA 1112 Series Elemental Analyzer, and a Conflow III open split interface (Thermo Finnigan, Bremen, Germany) at the University of Eastern Finland. The $^{15}$N data were expressed as atom % $^{15}$N excess relative to the natural abundance $^{15}$N of NO$_3^-$ and NH$_4^+$ in the soils from the unlabeled plots.

Dried bulk soil samples were also analyzed for total N and $^{15}$N concentrations using the same EA-IRMS, and atom % $^{15}$N excess values were calculated). The reproducibility of 10 standard runs (EA-IRMS) was typically better than 0.5‰ (1σ, n=10).

### 2.2.4 Plant sampling and analyses

Aboveground parts of plant and roots were quantitatively sampled from the labelled VP plots. Aboveground parts of plants were cut at the soil surface level and classified into higher plants (e.g., *Betula nana, Ledum decumbens, Rubus chamaemorus,*

*Vaccinium uliginosum*) and lower plants (e.g., *Sphagnum, Dicranum sp.*). Roots were removed by hand and rinsed with water to wash off soil. Then the aboveground parts of plants and roots were oven dried in the field laboratory, weighed and stored until further processing at the University of Eastern Finland. There, the aboveground biomass and roots were milled to fine powder (RetschMM301, Haan, Germany) and the total N and $^{15}$N contents in shoot and root material were determined by the EA-IRMS system described above.

## 2.3 Calculations

### 2.3.1 Mass Balance

To assess $^{15}$N partitioning and losses in the evaluated pools during the sampling period, we determined a mass balance which consisted of calculating the recovery of applied $^{15}$N into the different components of the ecosystem (plants, soil and N$_2$O) for



each sampling point. We used N pool size estimates (area-based) and changes in $^{15}$N content of the individual components following the $^{15}$N addition. All calculations were done with at% excess values which were calculated by subtracting the natural abundance of each component (which was measured before the labelling started; approx. 0.3663 at% $^{15}$N) from the measured at%15N values.

The mass of $^{15}$N recovered in each ecosystem component was determined as follows:

The atom % $^{15}$N natural abundance of each component (plants, including higher and lower plants, soil and gases) was measured before adding the label and the average value used for the calculations (1σ, n=12; approx. 0.3663 at% $^{15}$N for all).

(i)    We calculated the $^{15}$N mass recovered per soil area (µg $^{15}$N cm$^{-2}$) for each sampling time (e.g., at 0, 1 h, 24 h and 3, 5,

9, 15, 24 days) in each component (plants, soil or N$_2$O; cumulative fluxes for $^{15}$N-N$_2$O)

(ii) Total $^{15}$N recovery was expressed as a sum of the total mass of $^{15}$N recovered in each compartment at a given time, which consisted of somewhat different pools for BP (Eq. 1) and VP (Eq. 2). Since negligible concentrations of N$_2$O were emitted from VP, this compartment was ignored in the final mass balance calculations, and as there were no plants on BP, this compartment was also excluded in BP total $^{15}$N recovery calculation.

For BP surfaces:

Total $^{15}$N recovered (µg $^{15}$N cm$^{-2}$) = $^{15}$N$_{recovered}$ Soil (µg $^{15}$N cm$^{-2}$) + $^{15}$N$_{recovered}$ N$_2$O (µg$^{15}$ N cm$^{-2}$)          (1)

For VP surfaces:

Total $^{15}$N recovered = $^{15}$N$_{recovered}$ Soil (µg $^{15}$N cm$^{-2}$) + $^{15}$N$_{recovered}$ Plants (µg $^{15}$N cm$^{-2}$)          (2)

where $^{15}$N$_{recovered}$ Plants = $^{15}$N$_{recovered}$ Higher Plants + $^{15}$N$_{recovered}$ Lower Plants + $^{15}$N$_{recovered}$ Roots          (3)

The relative $^{15}$N recovery in each component (Eq. 4) was calculated dividing the $^{15}$N mass recovered per soil area (µg $^{15}$N cm$^2$) of each component by the total label applied to each individual replicate (3 x soil surface type). Here we report average total $^{15}$N recovered from each surface type (BP and VP) and average relative $^{15}$N recovery for each ecosystem component of BP (plants, soil and N$_2$O fluxes) and VP (plants and soil) surfaces in percentage (%). Only data from T1 ($^{14}$NH$_4$$^{15}$NO$_3$) and T2 ($^{15}$NH$_4$$^{14}$NO$_3$) were used for the mass balance calculation (T3 = sum of T1 and T2, data not shown).

$^{15}$N recovery (%) = ($^{15}$N$_{comp}$ (µg $^{15}$N cm$^{-2}$) / total label applied (µg $^{15}$N cm$^{-2}$)) x 100          (4)

### 2.3.2 Source partitioning of N$_2$O emitted from the bare peat surfaces (BP)

To quantify the relative contribution of nitrification and denitrification to the overall N$_2$O fluxes from the BP surfaces we used single and double $^{15}$N- labeled ammonium nitrate method, previously introduced by Baggs *et al.,* (2003). The calculation was made individually by plot (n=3) and sampling point (n=7). The average of three plots by day after labeling and the average of

the entire sampling period of 24 days is reported here. The microbial sources of N$_2$O were calculated as follows:

(i) The $^{15}$N-N$_2$O emitted from T1 plots (labeled with $^{14}$NH$_4$$^{15}$NO$_3$) was assumed to represent the $^{15}$N-N$_2$O derived from denitrification (D) (Eq. 5):

N$_2$O $_D$ = $^{15}$N$_2$O $_{T1}$          (5)



(ii) To separate N$_2$O derived from denitrification after $^{15}$NH$_4^+$ is nitrified to $^{15}$NO$_3^-$ from plots with T2 ($^{15}$NH$_4$$^{14}$NO$_3$), we used $^{15}$N-N$_2$O emitted data from plots with T1 in relation to the enrichment of the substrate pool ($^{15}$NO$_3^-$ ) from the plots under T1 and T2 (Eq. 6):

$$N_2O_{\text{ D-T2}} = (^{15}N_2O_{\text{ T1}} / \,^{15}NO_3^-{}_{\text{ T1}}) \text{ x } ^{15}NO_3^-{}_{\text{ T2}} \tag{6}$$

(iii) The N$_2$O-N derived from nitrification was then calculated from the difference between the $^{15}$N-N$_2$O emitted from T3 plots (labeled with $^{15}$NH$_4$$^{15}$NO$_3$) and T1 plots (labeled with $^{14}$NH$_4$$^{15}$NO$_3$), and accounting for $^{15}$NH$_4^+$ being nitrified to $^{15}$NO$_3^-$ (Eq. 7):

$$N_2O_{\text{ N}} = ^{15}N_2O_{\text{ T3}} - N_2O_{\text{ D}} - N_2O_{\text{ D-T2}} \tag{7}$$

The total $^{15}$N-N$_2$O emission was calculated as the sum of N$_2$O derived from denitrification (N$_2$O $_D$ and N$_2$OD $_{D-T2}$) and N$_2$O

derived from nitrification (N$_2$O $_N$), which was used to calculate the percent contribution of each process. The methodology used assumed that: (1) there was no dissimilatory NO$_3^-$ reduction to NH$_4^+$ (nitrate ammonification) and re-mineralization as $^{15}$N-NH$_4^+$ and (2) when using highly enriched isotopic tracers the isotopic composition of the N$_2$O evolved is not significantly affected by fractionation.

### 2.3.3 Gross N turnover rates

The experimental setup allowed us also to calculate gross mineralization and gross nitrification rates for VP and BP surfaces *in situ* with the isotope pool dilution method (Kirkham and Bartholomew 1954). We applied the pool dilution method in this study *in situ*, coupled with our virtual core technique and following the protocol suggested by Rütting *et al.* (2011).

The gross N transformation rates were calculated from T1 ($^{15}$N-NO$_3^-$; nitrification) and T2 ($^{15}$N-NH$_4^+$; ammonification) and were estimated only between 24 and 72 hours after labeling. This was done because (1) longer time periods (> 72 hours) after

labelling pose risks in erogenous calculations due to recycling of the label (Braun *et al.*, 2018). (2) the first time point of measurement (between 1 hour and 1 day after label application) could not be included in the calculations since negative rates were observed most likely because the label was not yet evenly distributed in the soil. The Eq. 8 and Eq. 9 of Kirkham and Bartholomew (1954) were used for the estimation of the gross mineralization/nitrification rate (*m*) and the gross NH$_4^+$/NO$_3^-$ consumption rate (*c*) (where m # c):

$$m = \frac{(M_0 - M_1)}{t} \text{ x } \frac{\log\left(\frac{H_0 M_1}{H_1 M_0}\right)}{\log\left(\frac{M_0}{M_1}\right)} \tag{8}$$

$$c = \frac{M_0 - M_1}{t} \text{ x } \frac{\log\left(\frac{H_0}{H_1}\right)}{\log\left(\frac{M_0}{M_1}\right)} \tag{9}$$

where M$_0$= initial $^{14+15}$N pool; M$_1$ = $^{14+15}$N pool at time 1, H$_o$= initial $^{15}$N$_{excess}$ pool, H$_1$ = $^{15}$N$_{excess}$ pool at time 1; t= time. All in µg N g$^{-1}$ dry soil.

Kirkham and Bartholomew (1954) methodology rely on the assumptions: (1) mineralization and immobilization rates remain

constant during the interval between successive measurements, (2) the ratio $^{15}$N/$^{14}$N in the efflux is in proportion to that of the labelled pool, and (3) immobilized labeled N is not remobilized during the experimental period (as mentioned above).





### 2.4 Statistical Analyses

Data was first tested for normal distribution using the normality test available in the Sigma Plot software (Systat, San Jose, CA). Since most of the data was not normally distributed, Kruskal-Wallis test was used to determine the significance of the experimental factors (surface type, $^{15}$N treatment, soil properties, air temperature) on $N_2O$ emissions. The Kruskal-Wallis test

was followed by Mann-Whitney pairwise test significant difference in the $^{15}$N recovery between treatments ($^{15}$N-NO$_3$ and $^{15}$N-NH$_4$) in each compartment (soil, plants and $N_2O$). Correlations of the $N_2O$ fluxes (labeled and unlabeled plots) with environmental variables, soil physical and chemical properties were tested using Spearman correlation analysis using IBM SPSS statistics software (version 23.0) and JMP®, Version Pro 14. SAS Institute Inc.

### 3 Results

### 3.1 Physicochemical characteristics of the soils

Physicochemical characteristics of BP and VP surfaces are summarized in Table 1. BP surfaces had higher bulk density than the VP surfaces, and particularly higher N content, resulting in much lower C/N ratios in BP as compared to VP. Both soils had similar low pH (mean = 3.4 ± 0.3). Water content was highly variable but on average higher in BP, with WFPS values ranging from 42% to 81% (mean 67 ± 5 %). In VP surfaces WFPS values ranged from 10 % to 57%, with a mean of 30 ± 7%.

In BP, the nitrate and ammonium contents (60 ± 11 mg N Kg-1 dry soil and 116 ± 39 mg N Kg-1 dry soil, respectively) were higher than in VP (11 ± 4 mg N Kg$^{-1}$ dry soil and 35 ± 6 mg N Kg$^{-1}$ dry soil, respectively) (all values are mean ± SE).

### 3.2 $N_2O$ emissions

Total $N_2O$ and $^{15}$N-$N_2O$ fluxes followed approximately similar seasonal patterns across all BP plots, with the highest $N_2O$ fluxes measured between day of the year (DOY) 210 and 215 (between three and nine days after $^{15}$N application) (Figure 1

and 3c). This peak in $N_2O$ fluxes was observed when temperatures of air ($\sim$ 18°C) and topsoil (5cm; 13°C) were highest. Bare peat surfaces showed net $N_2O$ release throughout the experimental period, ranging between 0.1 and 31.8 mg $N_2O$ m$^{-2}$ d$^{-1}$ (mean 9.8 ± 1.8 mg $N_2O$ m$^{-2}$ d$^{-1}$, n= 44) and were on average about 3 times higher than those from adjacent, non-labelled bare peat areas (mean 3.2 ± 0.5 mg $N_2O$ m$^{-2}$ d$^{-1}$, n= 34; Figure 1). The highest $^{15}$N-$N_2O$ flux in BP was measured from the T3 treatment ($^{15}$NH$_4$$^{15}$NO$_3$) ($p < 0.05$) (Figure 1). The $N_2O$ fluxes from the VP surfaces were low throughout the sampling period

and showed frequent uptake of $N_2O$ (negative fluxes). The $N_2O$ fluxes in VP from the $^{15}$N labelled plots ranged from -1.6 to 4.3 mg $N_2O$ m$^{-2}$ d$^{-1}$ (mean -0.02 ± 0.14 mg $N_2O$ m$^{-2}$ d$^{-1}$; n= 55) and were not significantly different from zero, and not significantly different from adjacent non-labelled VP areas (data not shown).

$N_2O$ fluxes correlated positively with air T ($R^2$= 0.357; $p < 0.005$), NH$_4$$^+$ concentration in soil ($R^2$=0.423; $p < 0.001$) and CO$_2$ fluxes ($R^2$=0.399; $p < 0.005$). $^{15}$N$_2O$ fluxes from labelled plots showed similar positive correlation with air T as $N_2O$ fluxes

($R^2$ = 0.391, p < 0.005).

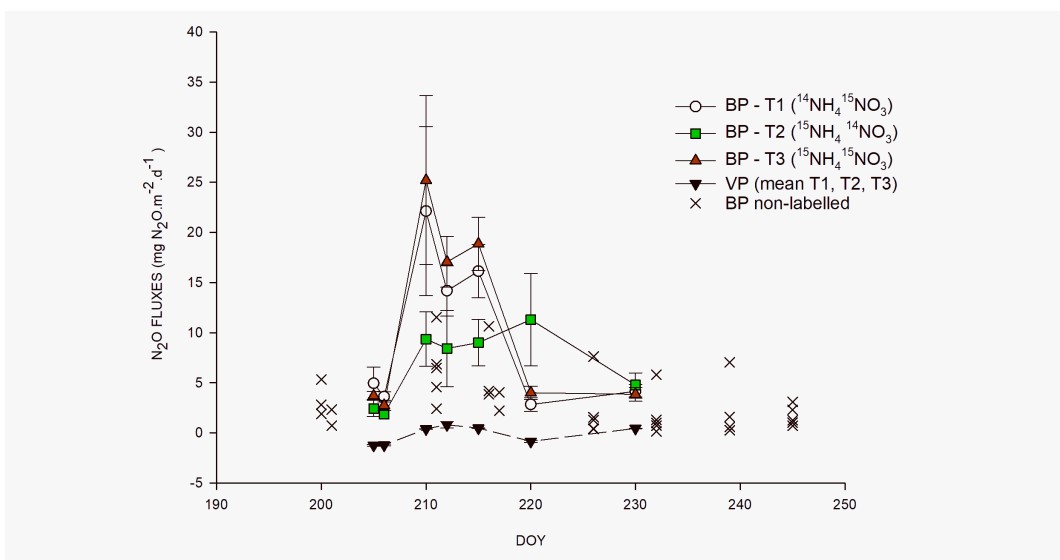

**Figure 1.** Total $N_2O$ fluxes from the bare peat soils (BP) from labelled (color) and no-label plots (x) and vegetated peat soils (VP) from labelled plots. For VP soils, mean $N_2O$ flux of three treatments is shown. For labelled plots $N_2O$ fluxes are mean values ± SE (n = 3). DOY= day of the year. T1 = treatment 1 ($^{14}NH_4^{15}NO_3$); T2 = treatment 1 ($^{15}NH_4^{14}NO_3$) and T3 = treatment 3 ($^{15}NH_4^{15}NO_3$).

**3.3 $^{15}N$ recovery**

The total amount of $^{15}N$ recovered in the soil, vegetation and $N_2O$ were calculated from treatments T1 ($^{15}N$-$NO_3^-$) and T2 ($^{15}N$-$NH_4^+$). In general, the total recovery was close to 100% for the first 24 hours after labelling and gradually decreased after 24 days to 42% for BP and 75% for VP (Figure 1S). Immediately after labelling (24h), 92% (VP) and 100% (BP) of the applied $^{15}N$ was recovered in the bulk peat soil. By the end of the experiment, still most of the label across VP and BP was found in

the bulk peat soil in both treatments, as shown in the relative proportion of each component (Figure 2).

In VP the $^{15}N$ recovered in plants 24 days after labeling was on average 6 ± 2 % (n = 42) of the total label recovered for both T1 ($^{15}N$-$NO_3^-$) and T2 ($^{15}N$-$NH_4^+$), with no significant difference between the T1 and T2. The relative proportion of the label recovered in vegetation did not show a consistent trend over the experimental period varying from 1 to 9% (Figure 2c-d). Most of the $^{15}N$ in vegetation was retained in mosses and lichens (3 to 4 %), followed by roots of higher plants (2%) and

aboveground parts of higher plants (0.2%) (Figure S2).

Since the $N_2O$ emissions from vegetated peat were negligible (Figure 1), the $^{15}N$-enrichment of this $N_2O$ was not determined for the VP surfaces. In BP, the $^{15}N$-enrichment of the $N_2O$ was detected after three days of treatment, with cumulative increases with time (Figure 2a-b). The maximum $^{15}N$-recovery in $N_2O$ from BP surfaces was observed toward the end of the experiment (Day 24) from T1 ($^{15}N$-$NO_3$) (24 ± 9 %; n = 3). On average, the label recovered in $^{15}N$-$N_2O$ was higher in the T1 ($^{15}N$-$NO_3^-$)

plots (13 ± 2 %; n = 3) compared to T2 ($^{15}N$-$NH_4^+$) plots (6 ± 1%; n = 3, $p < 0.05$). The maximum relative amount of $^{15}N$-recovery in $N_2O$ in BP surfaces was about 3.5 and 1.5 times higher than the maximum $^{15}N$ recovery in plants in VP for the treatments T1 and T2, respectively.





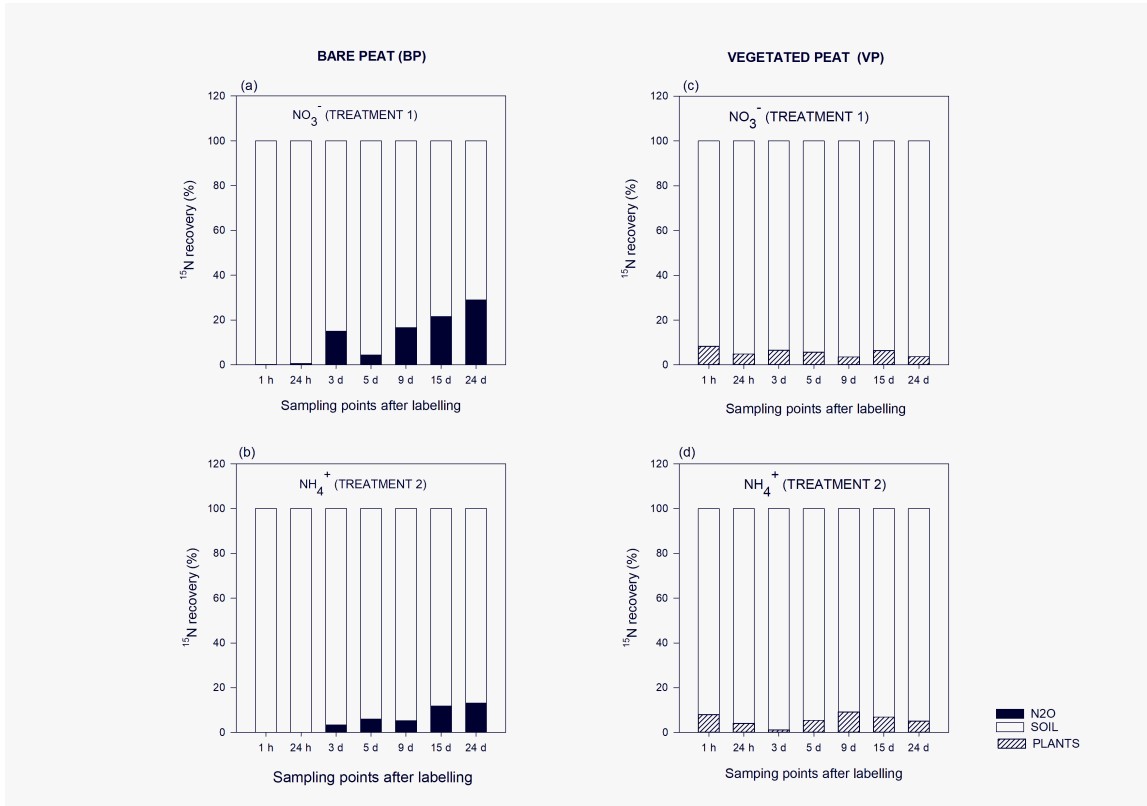

**Figure 2**. Relative distribution of the $^{15}N$ label recovered from the bare peat and vegetated peat soils for treatment 1 ($NH_4{}^{15}NO_3$) and treatment 2 ($^{15}NH_4NO_3$).

### 3.4 $^{15}N$-concentrations of inorganic N pools and $N_2O$, and microbial sources of $N_2O$ emitted from bare permafrost

5    **peatlands**

In the labeling treatment T1 ($^{15}N$-$NO_3{}^-$), the highest $^{15}N$-$NO_3{}^-$ concentration was measured one day after labeling ($0.8 \pm 0.5$ mg $^{15}N$-$NO_3{}^-$ Kg$^{-1}$ of dry soil) (Fig. 3b). In the same treatment, the $^{15}N$ concentration of the $NH_4{}^+$ pool was negligible during the 24 days of experiment ($\sim 0.1$ mg $^{15}N$-$NH_4{}^{+}$ Kg$^{-1}$ dry soil) indicating there was no reduction of nitrate to ammonium.

In the treatment T2 ($^{15}N$-$NH_4{}^+$), the concentration of $^{15}N$-$NH_4{}^+$ decreased generally exponentially over time (Figure 3a). In

10    the same treatment, $^{15}NO_3{}^-$ gradually increased during the first nine days after labeling and thereafter decreased after 24 days. In the treatment T3 ($^{15}NH_4{}^{15}NO_3$), the $^{15}N$-$NO_3$ concentration of soil showed a similar trend as in T1 but with higher $^{15}N$-concentration (Figure 3b). The $^{15}NH_4{}^+$ pool in T3 showed similar trend to the $^{15}NH_4{}^+$ concentrations from T2, but $^{15}N$ concentration were at lower levels. In nearly all treatments, a second smaller peak was detected in $^{15}N$ concentrations of the added substrate on day 5, 9 or 15.

15    The $^{15}N$ concentration in $N_2O$ showed similar patterns for all treatments. The highest $^{15}N$-$N_2O$ flux ($7 \pm 3$ mg $^{15}N$-$N_2O$ m$^{-2}$ d$^{-1}$) was measured from the T3 on day three after labeling (Figure 3c). The same was true also for T1, but the $^{15}N$ level was lower ($3 \pm 1$ mg $^{15}N$-$N_2O$ m$^{-2}$ d$^{-1}$). For T2, the peak in $^{15}N_2O$ flux ($1.6 \pm 0.5$ mg $^{15}N$-$N_2O$ m$^{-2}$ d$^{-1}$) was lower and occurred later,





between day 3 and 5 after the application of the label. In all treatments, a second smaller peak in $^{15}N_2O$ was observed, but earlier in T1 and T3 (about 2 days) compared to T2. The $^{15}N_2O$ values correlated positively with the $^{15}NO_3^-$ values from all treatments ($R^2 = 0.5453$; $p < 0.05$, Figure S3), but not correlation between $^{15}NH_4^+$ and $^{15}N_2O$ was observed.

The results of the source partitioning of $N_2O$ emissions from BP surfaces (Figure 4) show that denitrification was the primary

5  source of $N_2O$ from BP surfaces contributing on average $79 \pm 6\%$ ($n = 21$) to the total $^{15}N\text{-}N_2O$ emissions. In T2 ($^{15}NH_4^{14}NO_3$), there was $^{15}N$ in $NO_3^-$ pool indicating that the applied $^{15}N\text{-}NH_4^+$ was nitrified and released as $^{15}N\text{-}N_2O$ in coupled nitrification-denitrification process. The relative contribution of ammonia oxidation to the overall $N_2O$ flux was ~20%. During the period of high $N_2O$ fluxes (3 days after $^{15}N$ application), the contribution of nitrification was particularly low. However, when $N_2O$ emissions were low toward the end of the experiment, nitrification reached a maximum contribution of 55%.




Biogeosciences Open Access
Discussions
EGU

**Figure 3**. Evolution of $^{15}N$ enrichment in (a-b) extractable inorganic N pools ($NH_4^+$ and $NO_3^-$) and (c) $N_2O$ emissions from bare peat (BP) soil during the 24-day of the experiment for all labelling treatments. T1 = treatment 1 ($^{14}NH_4^{15}NO_3$); T2 = treatment 1 ($^{15}NH_4^{14}NO_3$) and T3 = treatment 3 ($^{15}NH_4^{15}NO_3$). Day 0 = 1hour after labeling. Values are mean ± 1 SE (n = 3).

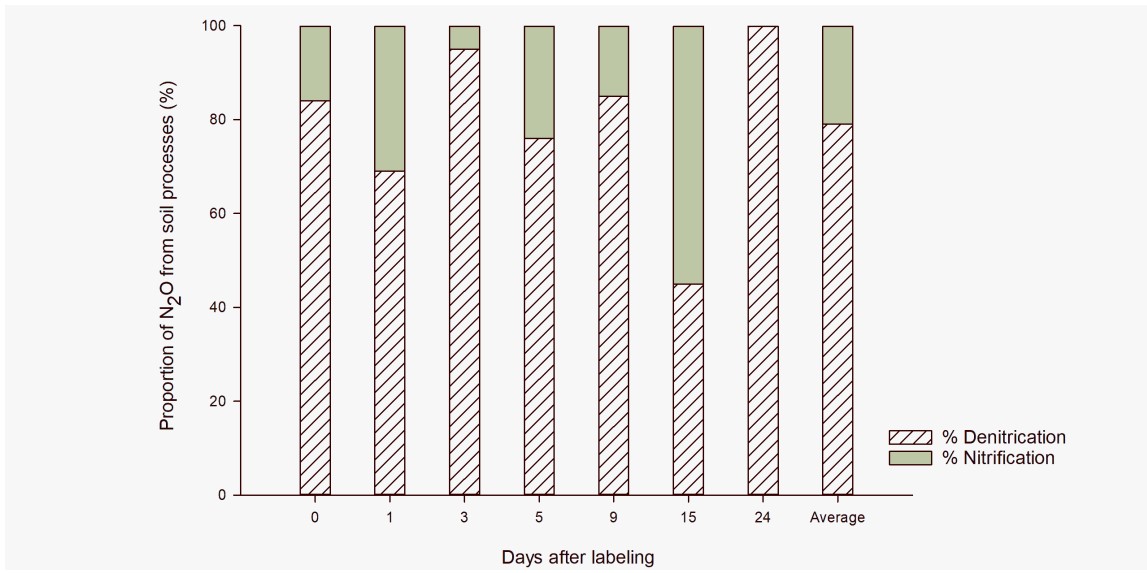

**Figure 4**. Proportion of $N_2O$ produced in the bare peat (BP) soil by denitrification and nitrification during the study period. The source partitioning was calculated using single and double $^{15}N$- labeled ammonium nitrate method (Baggs *et al.*, 2003). The source contribution is calculated from $^{15}N_2O$ emitted and $^{15}NO_3^-$ pool in the soil, in relation to the total amount of $^{15}N$ label applied to the soil (Wrage *et al.*, 2005). The calculation was done individually by plot (n=3) and sampling point (n=7) and the average of three plots is reported by day after labeling. The average of the entire sapling period is also shown.

### 3.5 Gross N turnover rates

The gross N transformation rates were calculated between 24h and 72h for both BP and VP soil surfaces (Table 2). Gross mineralization and nitrification rates for BP surface were $2.6 \pm 0.9$ µg cm$^{-3}$ d$^{-1}$ and $0.9 \pm 0.5$ µg cm$^{-3}$ d$^{-1}$ respectively, while for VP surface they were $0.1 \pm 0.1$ µg cm$^{-3}$ d$^{-1}$ and $0.010 \pm 0.003$ µg cm$^{-3}$ d$^{-1}$, respectively. Gross mineralization and nitrification rates in BP were 27 and 90 times higher than in VP (p < 0.01), respectively. For BP surfaces, gross mineralization rates were five times higher than gross nitrification rates. There were very low and similar $NH_4^+$ consumption rates detected between soil surfaces (*P*<0.05), while $NO_3^-$ consumption only took place in VP surface and not in BP, being the highest turnover rate in VP. Calculated on a soil weight basis (in µg N g$^{-1}$ d$^{-1}$), gross mineralization and gross nitrification rates, respectively, were $9.9 \pm 3.5$ and $3.2 \pm 1.9$ µg N g$^{-1}$ d$^{-1}$ in BP and $1.3 \pm 1.0$ and $0.2 \pm 0.1$ µg N g$^{-1}$ d$^{-1}$ in VP (see supplementary material, Table S1).





**Table 2**. Gross N rates from bare peat (BP) and vegetated peat (VP) surfaces calculated from mineral N pools in the soil.

| Surface type | Mineralization (µg N cm$^{-3}$ d$^{-1}$) | NH$_4^+$ consumption (µg N cm$^{-3}$ d$^{-1}$) | Nitrification (µg N cm$^{-3}$ d$^{-1}$) | NO$_3$- consumption (µg N cm$^{-3}$ d$^{-1}$) |
|---|---|---|---|---|
| BP | 2.7 ± 0.9[a] | 0.07 ± 0.02 | 0.9 ± 0.5[a] | 0.00 |
| VP | 0.1 ± 0.1[b] | 0.03 ± 0.01 | 0.010 ± 0.003[b] | 0.2 ± 0.1 |

Values are mean ± 1 S.E; n=3
Different letters indicate statistically significant differences between the surface types ($P$ <0.05).

## 4 Discussion

### 4.1 N$_2$O flux rates from Bare and Vegetated Peat soils

Similar to previous studies at the study site, the N$_2$O fluxes from the BP surfaces (mean 3.2 ± 0.5 mg N$_2$O m$^{-2}$ d$^{-1}$) were higher

than from VP surfaces, from which N$_2$O fluxes were negligible throughout the sampling period (mean -0.02 ± 0.14 mg N$_2$O m$^{-2}$ d$^{-1}$). The emission rates were highest at warmest air temperatures (R$^2$= 0.357; $p$ < 0.005). Nitrous oxide fluxes from BP are comparable to the emissions generally reported from drained boreal peatlands used for agriculture (0.1 – 15.1 mg N$_2$O m$^2$. d$^{-1}$) (Maljanen *et al.,* 2010) and from tropical forests (0.09 – 2.5 mg N$_2$O m$^{-2}$ d$^{-1}$). Tropical forests are among the most important natural terrestrial ecosystems in terms of N$_2$O emissions (Werner *et al.,* 2007), while it was generally assumed that

N$_2$O emissions from Arctic soils are negligible. Contrary to this general pattern, the results here confirm the earlier findings that there are surfaces in the Arctic, namely bare peat soils on permafrost peatlands, with the potential to emit substantial amounts of N$_2$O (Repo *et al.,* 2009; Marushchak *et al.,* 2011; Voigt *et al.,* 2017a).

The bulk N$_2$O fluxes from the $^{15}$N-labelling subplots were on average 3 times higher than those from adjacent, non-labelled bare peat areas. The concentration of inorganic N was at most doubled by adding labelled NO$_3^-$ and/or NH$_4^+$, but the final

nutrient content never exceeded maximum content of native NO$_3^-$ or NH$_4^+$ observed in the soil (data not shown). The bulk N$_2$O fluxes from the labelled plots (~10 mg N$_2$O m$^{-2}$. d$^{-1}$) were still within the range of N$_2$O fluxes observed in previous years from BP surfaces (1.9 – 31 mg N$_2$O.m$^{-2}$d$^{-1}$) (Repo *et al.,* 2009; Marushchak *et al.,* 2011). The differences in the N$_2$O fluxes from BP labelled and non-labelled plots could be also attributed to the natural spatial variation in the N$_2$O fluxes within the BP surfaces, which can be large even on small spatial scales (< 1m, personal observation. data not shown). The N$_2$O emissions

from labelled and non-labelled plots had similar responses to changes in temperature (R$^2$ = 0.391, p < 0.005), which was likely the major factor controlling the temporal variation in the N$_2$O fluxes from BP surfaces during the study period. Even if some stimulation occurred, there likely was no change in the relative contribution of various processes underlying the N$_2$O emissions because BP surfaces were not N limited during the study period (see discussion below).

### 4.2 Gross mineralization and nitrification rates from Bare and Vegetated Peat soils

Gross mineralization and nitrification rates were higher in BP than in VP surfaces (Table 2), and thus the lower C/N ratio under BP surfaces together with higher WFPS seemed to favor N turnover supporting hypothesis three of this study. In





addition, higher consumption of $NO_3^-$ in VP, and the low or lack of consumption of it in BP, suggested that the microbial demands are met in BP surfaces while in VP microbes are N deprived, and any N available would be immobilized. This suggestion agrees with the findings of Diáková *et al.,* (2016) where significant higher net N mineralization rates were observed in BP surfaces compared to VP, indicating that microbial communities in BP surfaces had a surplus of available N. It has to

be noted that $NO_3$ consumption exceeded gross $NO_3$ production (nitrification) in VP, which cannot be easily explained by nutrient dynamics of natural soils receiving no fertilizer input. Likely, this was due to the small $^{15}NO_3$ additions, however, the rates still indicate sever N limitation in VP. Further, we noted that in BP, $NH_4^+$ consumption was about 10-times lower than gross nitrification, which could indicate that heterotrophic nitrification takes place (Figueiredo *et al.,* 2016). Nitrogen turnover processes need further investigations in these soils.

Gross N mineralization rates in BP surfaces ($2.7 \pm 0.9$ µg N cm$^{-3}$ d$^{-1}$) were higher compared to the gross mineralization rates reported for boreal peatlands (1 to 2 µg N cm$^{-3}$ d$^{-1}$; Westbrook & Devito, 2004) and within range reported for mineral tundra soils (mineral and organic horizon) (0.1 to 9 µg N cm$^{-3}$ d$^{-1}$) (Biasi *et al.,* 2005, Meyer *et al.,* 2006; Buckeridge *et al.*, 2007; Marushchak *et al.,* 2011) and organic layers of spruce forest soil (1 to 4 µg N cm$^{-3}$ d$^{-1}$) (Brüggemann *et al.,* 2005; Zeller *et al.,* 2008). The gross N mineralization and nitrification rates of BP expressed per g dry weight (Table Sx; 9.9 and 3.2 µg N g$^-$

$^1$ d$^{-1}$) were also comparable to rates found in boreal, temperate and tropical soils (Booth *et al.,* 2005), and were in line with results from previous studies from Arctic ecosystems (e.g., Kaiser *et al.,* 2007; Wild *et al.,* 2015). However, these relatively high N turnover rates contradict the general idea that organic N cycling dominates in cold ecosystems and mineral N cycling is of low importance (Schimel *et al.,* 2004). Instead, it seems that gross N mineralization rates and gross nitrification rates can be high in arctic and sub-arctic ecosystems, if conditions are favorable (e.g., low C/N ratio, high %N, suitable water content;

Booth *et al.,* 2005). In VP, however, gross N turnover rates were at low end of rates reported from other ecosystems (Booth et al., 2005), but still measurable. Likely, the differences in mineral N cycling explain at least partly also the differences in $N_2O$ fluxes found between BP and VP, as discussed below.

### 4.3 Fate of mineral N and factors affecting $N_2O$ production

The total recovery of applied $^{15}N$ within 24 hours for both studied surface types was close to 100%. However, this % decreased

during the course of the post-application sampling in both VP and BP surfaces, which might be a consequence of lateral and vertical movement of N forms, particularly $^{15}NO_3^-$, in the soils, including downward leaching (Clough *et al.,* 2001), and possibly also of increasing importance of NO fluxes and $N_2$ production which we did not measure here. Downward leaching is a possible explanation for the observed decrease in total recovery of the label, since the total recovery of $^{15}N$ was higher in VP than in BP surfaces during the 24 days of experiment (~79% vs. ~62%, respectively) because of plant N uptake and

microbial immobilization. It is also possible that the $^{15}N$ might have increasingly accumulated as $^{15}N-N_2O$ and $^{15}N-N_2$ in soil solution or in gas filled pores in BP. Soil gas concentrations of $N_2O$ can reach high concentrations (up to 4500 ppb) particularly in BP where $N_2O$ is mainly produced in these permafrost peatlands (Gil *et al.,* 2017). However, since still more than 60% and 80% of $^{15}N$ was recovered in VP and BP on average, respectively, we accounted for all the major sinks of $NO_3^-$ and $NH_4^+$ in both soils throughout the 24 days of incubation.

For VP and BP, the largest relative contribution of $^{15}N$ label was observed in bulk peat (71-92% of total $^{15}N$ recovered) after 24 days of experiment, comprising physically adsorbed, dissolved, chemically or electro-chemically fixed and microbially



immobilized $^{15}$N. Peatlands are known to be able to efficiently retain nutrients to deal with low N inputs, which has given them important evolutionary advantages and ecological functions as nutrient buffers (Vikman et al, 200). Recovery of $^{15}$N in bulk peat was higher for $^{15}$NH$_4^+$ than for $^{15}$NO$_3^-$ (Figure 2a and 2b), suggesting that fixation of nutrients to SOM is one of the main reasons for the high retention of $^{15}$N, since soil particles are negatively charged (Schlesinger, 1997) and since fixation

capacity is high under acidic conditions (Huber, Oberhauser & Kreutzer, 2002). This is supported by other studies that have found evidence for similarly high fixation of nutrients, particularly $^{15}$NH$_4$, to organic peat material (e.g., Munchmeyer *et al*., 2000). Microbial immobilization likely is another reason in VP, since immobilization rates were high in VP where C/N ratio is comparatively high (C/N ~62) and %N is low (0.8%) and where content of inorganic N is constantly low throughout the season (Table 1) (Marushchak *et al.,* 2011; Voigt *et al.,* 2017a). This was also confirmed by our results on NO$_3^-$ consumption

obtained from the pool dilution approach (Table 2 and see discussion below). Rapid uptake of $^{15}$N by microbes in soils with low N availability from arctic and sub-arctic ecosystems have been documented during the first days after the addition of label (Nordin, Schmidt & Shaver, 2004; Sørensen *et al.,* 2008).

It has been shown that in the short term, plants compete poorly for available N, but this competition depends on the season and many other factors (Grogan & Jonasson, 2003; Nordin, Schmidt & Shaver, 2004). In our 3-weeks period study, the average

$^{15}$N uptake by the plants (vascular plants + mosses) of ~6% was of the same order of magnitude as in reports from other arctic ecosystems (1 to 5 %) for different sampling length (from 4 h up to 12 weeks) (Grogan & Jonasson, 2003; Nordin, Schmidt & Shaver, 2004). The fact that $^{15}$N in plants did not constantly increase in our experiment (Figure S2), suggests that the label, once incorporated into the soil, is only slowly released in plant-available forms as suggested also by others (Sorensen *et al.,* 2008). Generally, following the soil most of the label was recovered in mosses (3-4%), followed by roots (~1 to 3%) and

aboveground vascular plant parts (< 1%) in VP. The relatively large difference in $^{15}$N observed between mosses and vascular plants might be related to the difference in their mechanism for nutrient acquisition. Mosses derive N principally from atmospheric deposition (e.g., wet deposition) but also from soil N, and their nutrient acquisition is thought to relate to the pattern of water uptake (Ayres *et al.,* 2006) and is passive. Since the $^{15}$N tracers were added in water solution this should have facilitated the uptake of the $^{15}$N label by the mosses in VP surfaces, which penetrate the upper soil column where the label

was injected.

It has been shown that plants from different ecosystems, including Arctic ecosystems, can show N uptake flexibility between forms of N (organic N, NO$_3^-$, NH$_4^+$) based on environmental conditions and species competition (McKane *et al.,* 2002; Gao *et al.*, 2020). In our study, there was no difference in the plant uptake of $^{15}$N-NO$_3^-$ and $^{15}$N-NH$_4^+$ in VP surfaces. The $^{15}$N in plants was determined for the bulk and not for individual species and is possible that discrimination between the N forms based on species-specific preferences could take place (Gao *et al.*, 2020).

In BP, where plants were absent, 24% of the applied $^{15}$N was detected in the cumulative N$_2$O emission at the end of the experiment. The recovery of the label in N$_2$O in BP is thus up to threefold larger than the relative portion of label observed in plants in VP (maximum value ~ 9%). This confirms our second hypothesis, that a higher proportion of the added $^{15}$N is released in gaseous form in BP than taken up by plants and immobilized in VP. It suggests that competition for N is an

important regulator of N$_2$O in these peatlands, and that plants control to some extent the emissions of this strong greenhouse gas. This has been observed before for restored boreal peatland with various levels of nitrate addition and plant coverage and drained forested peatland, where presence of roots halved N$_2$O emission (Silvan *et al.,* 2005; Holz *et al.,* 2016). It is likewise supported by recent results from a mesocosms study which show that presence of vegetation limits N$_2$O emissions from tundra





soil by ~90% (Voigt *et al.,* 2017b). In BP, on the other hand, in the absence of plants excess mineral N was highly available for microbial $N_2O$ production processes (e.g., nitrification, denitrification) (Schimel & Bennett*,* 2004) and microbes are not N limited since there is always enough amounts of $NO_3^-$ and $NH_4^+$ available, and there is no competition between plants and microbes. The differences in $N_2O$ emissions between BP and VP are further a direct consequence of variable production rates

of mineral N forms, with VP having much lower gross N mineralization and nitrification rates than BP likely due to higher C/N ratios of the soils. Another, important factor limiting $N_2O$ production in VP is likely the low WFPS (29 % ± 1 %; Table 1) and thus high aeration status of peat in VP which deceases denitrification potential (Firestone and Davidson, 1989.). The higher WFPS in BP, on the other hand, creates ideal conditions for mineralization, nitrification and denitrification to take place (Liimatainen *et al.,* 2018). To conclude, in VP with low C/N ratio and high aeration status, $N_2O$ production is limited

by low mineralization and nitrification rates together with plant N uptake and immobilization of N. We thus find strong support for hypothesis 2 in this study.

**4.4 Microbial source of $N_2O$ emitted from the bare peat surfaces**

The source partitioning approach suggests a general dominance of denitrification processes (~79%) as a source of $N_2O$ in BP surfaces. The results of the source partitioning approach are also corroborated by the higher $^{15}N$-$N_2O$ flux levels after

application of $^{15}N$-$NO_3$ compared to application of $^{15}N$-$NH_4$. The soil properties and N dynamics hint also at denitrification pathways being dominant in BP surfaces, where the high $NO_3^-$ content and the inter-mediate to high soil moisture conditions cause high $N_2O$ emissions via denitrification, as also suggested previously (Repo *et al.,* 2009; Palmer *et al.,* 2011; Marushchak *et al.,* 2011). Hypothesis one of this study was thus supported by our labelling study and source partitioning approach. On a side note, we cannot clearly explain the second peak which we found in $^{15}N_2O$ and several inorganic $^{15}N$ pools in BP, but this

could be due to immobilization and later re-cycling of added $^{15}N$ by microbes (Braun *et al.,* 2018).
Denitrification was the dominant microbial process responsible for $N_2O$ emissions in BP. This is supported by findings of Palmer *et al.* (2011), who detected a high number of functional genes involved in denitrification in these soils and high potential for denitrification. Few, highly specialized taxa, mostly belonging to the family of Burkholderiaceae (co-occurring with Rhodanobacter sp.*)*, seem to be responsible for most of the denitrification occurring in these acidic soils using acetate as

their energy source (Hetz & Horn, 2021). Despite the clear dominance of denitrification, the relative contribution of total nitrification to the $N_2O$ emissions from the BP surfaces (~20%) was still significant and could be particularly important during drier summers (low soil water content) and at the end of the growing season when the $N_2O$ emissions are generally lower, as shown here and also in Gil *et al.* (2017). In 2010, we found evidence for nitrification derived $N_2O$ via $^{15}N$ natural abundance approaches in an exceptionally dry year in Seida, where WFPS of BP was almost 20% less than in 2010 (this study) and $N_2O$

emissions were much lower (Gil et al., 2017). This corroborates that moisture is a key $N_2O$ emission driver for the prevalence of nitrification or denitrification to occur. Nitrifier denitrification can be, however, ruled out as a possible source of $N_2O$ in these soils since we know now that in Seida peat ammonia oxidizing archaea (AOA) are responsible for ammonia oxidation and ammonia oxidizing bacteria (AOB) are lacking there (Siljanen *et al*., 2019; Hetz & Horn, 2021). AOA are not capable for denitrification (anaerobic) in contrast to AOB. Thus, if $N_2O$ nitrification (ammonium oxidation) is the source for $N_2O$, it

is not produced in nitrifier denitrification. Nitrite produced by AOA could allow some chemical production of $N_2O$ (chemical denitrification) when nitrite reacts with soil organic matter in acidic conditions like prevailing here (Kappelmeyer *et al.,* 2003).





Since physical and chemical conditions in the studied permafrost peat surfaces are favorable for both nitrification and denitrification, it is possible that adjacent aerobic and anaerobic microhabitats enabled both ammonia oxidation and denitrification to occur and produce $N_2O$. Furthermore, nitrate/nitrite from nitrification is used as an electron acceptor in denitrification and $N_2O$ is produced through a coupled nitrification-denitrification process in this C rich soil, as supported by

our data. Overall, our results support the assumption that denitrification is mostly responsible for the high $N_2O$ emissions from BP surfaces during the study period in 2010, where $N_2O$ emissions were high and temperature and precipitation were within the range of mean values recorded for the site. Nitrification is, on the other hand, taking place in BP surfaces and is a key process involved in $N_2O$ production in these soils directly and indirectly through the production of the substrate ($NO_3^-$) for denitrification (Siljanen *et al,* 2019). This implies that moisture conditions are key for controlling the main production

pathway of $N_2O$ release from permafrost soils.  Since the balance between nitrification and denitrification in soils influences $N_2O$ emission strength, and a shift towards denitrification causes higher $N_2O$ emissions, increased soil water content as it is predicted for Alaska and likely other permafrost regions (Douglas *et al.,* 2020) might stimulate $N_2O$ emissions from sites with native soil characteristics favorable for $N_2O$.

**5 Conclusions**

The $N_2O$ emission rates from the BP surfaces (mean 3 mg $N_2O$ $m^{-2}$ $d^{-1}$) were high, as hypothesized, while $N_2O$ emissions from VP were negligible throughout the sampling period. In VP, $N_2O$ production was limited by the low inorganic N content and low delivery of N, as opposed to BP surfaces. For both, VP and BP, most of [15]N label was observed in the peat soil after 24 days of experiment, followed by $N_2O$ in BP surfaces and plants in VP. The recovery of the label was larger in $N_2O$ in BP than in plants in VP. This suggests that the competition for nutrients, the generally lower availability of nutrients, as compared

to BP because of high C/N, and low mineralization and nitrification rates limits the $N_2O$ release in VP. In addition, low bulk density in VP (high porosity) and lower water content indicates more aerobic conditions, compare to BP surfaces, limiting $N_2O$ production by denitrification in VP.

The source partitioning of $N_2O$ from BP surfaces supported the assumption that denitrification is the dominant process behind the high $N_2O$ emissions from BP during the study period in 2010. Further, it showed that also nitrifying processes are taking

place and emitting some $N_2O$ in BP surfaces. Although denitrification is the dominant source of $N_2O$ in this system, nitrification is a key process involved in $N_2O$ production in these soils directly and indirectly through the production of the substrate ($NO_3^-$) for denitrification.  With future warming, increased rainfall, permafrost thaw and collapse more anaerobic conditions might prevail, which might increasingly emit $N_2O$. These processes need to be considered in global N budgets and in N cycling models for permafrost regions which are currently being developed.


Our results also suggested that VP surfaces are N limited while the opposite is true for BP surfaces. Thus, in BP surfaces, in the absence of plants, any excess of N is completely available for microbes and together with favorable environmental conditions (intermediate to higher soil moistures, low C/N, high bulk density) results in high $N_2O$ emission from BP surfaces. This might be also critical for future climate feedbacks of Arctic ecosystems, since thermokarst resulting from permafrost

thawing often causes disturbance of the vegetation cover and increase in moisture content, spiraling the risks for $N_2O$ release.



On the other hand, overall trends towards increasing plant growth in a warming Arctic might slow down N$_2$O release in the long-term. The net effect of all these changes on N$_2$O emissions from permafrost regions are currently not known but need to be the focus of future studies.

## 6 Data availability

Most of the data is provided in the figures and tables in the paper and any additional data may be obtained from J. Gil (email: gillugoj@msu.edu).

## 7 Author contribution

C.B., E.M.B., T.R. and P.J.M. designed the study; J.G., M.E.M., C.B., T.T. and A.N., conducted the field work; D.K. and A.N. provided access to and expertise on the study sites and supported the project with field logistics, J.G. and C.B. conducted

laboratory analysis and data processing; J.G. wrote the first version of the manuscript with contribution from C.B. and T.P., after which all co-authors provided input on manuscript text, figures, and discussion of scientific content.

## 8 Competing interests

The authors declare that they have no conflict of interest.

## 9 Acknowledgment

This study was financially supported by the Academy of Finland, project CryoN2010-2014 (Mechanisms underlying large N$_2$O emissions from cryoturbated peat surfaces in tundra; decision Nr. 132045). We acknowledge further funding from the European Union 7th Framework Program under project Page 21 (contract Nr. GA282700); Academy of Finland, project NOCA (decision Nr. 314630); EMPIR project SIRS (Metrology for stable isotope reference standards, project number 16ENV06 SIRS), and (DEFROST)-Nordic Centre of Excellence Program (Impacts of changing cryosphere – depicting

ecosystem-climate feedbacks from permafrost, snow and ice). We are also grateful to the consortium of the ESF supported project CryoCARB for productive discussions. J. Gil acknowledges salary funding from University of Eastern Finland during 2014 and the Great Lakes Bioenergy Research Center, U.S. Department of Energy, Award Number DE-SC0018409 during 2018-2021 and personal grant awarded by Finnish Cultural Foundation (2015) and Academy of Finland/Vilho, Yrjö ja Kalle Väisälän Foundation (2016). We also thank Jouko Pokela, Slava Hozainov, Igor Marushchak, Natalia Kaneva and Henry

Mora for their invaluable help during the fieldwork. The authors would like to thank the anonymous reviewers for their insightful comments and suggestions that have contributed to improve our manuscript significantly.

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
