# Peer review of "Sources of nitrous oxide and fate of mineral nitrogen in sub-Arctic permafrost peat soils"

_Biogeosciences, 2021_

## Author Comment (AC1)

**Comment on bg-2021-228**

**Anonymous Referee #2**

[Figure]

**Figure S4.** 15N at% excess (APE) of (a) NH4+ (b) NO3- and (c) N2O during the sampling period for all three treatments applied in BP surfaces. (Values are mean ± S.E, n=3). Only data from treatment 1 (T1 = 15N-NO3-) and treatment 2 (T2 = 15- NH4+) was used in the calculations of the gross N transformation rates.

---

## Author Comment (AC2)

**Comment on bg-2021-228**

**Anonymous Referee #1**

**Figure S3**. Change in natural logarithmic of 15N at% of (a - b) NH4+ an (c - d) NO3- in BP and VP soil over total study period.

**Figure S4.** 15N at% excess (APE) of (a) NH4+ (b) NO3- and (c) N2O during the sampling period for all three treatments applied in BP surfaces. (Values are mean ± S.E, n=3). Only data from treatment 1 (T1 = 15N-NO3-) and treatment 2 (T2 = 15- NH4+) was used in the calculations of the gross N transformation rates.

Table S1. Gross N process rates from bare peat surfaces (BP)

| (a) | | µg N cm-3 d-1 Mineralisation | µg N cm-3 d-1 NH4 consumption | µg N cm-3 d-1 Nitrification | µg N cm-3 d-1 NO3 conssumption |
|---|---|---|---|---|---|
| 24 hours | average | 8,76 | 8,10 | 0,30 | 0,23 |
| | std | 2,36 | 1,93 | 0,42 | 0,84 |
| | se | 1,36 | 1,11 | 0,24 | 0,48 |
| 72 hours | average | 3,31 | 3,50 | 0,87 | 0,92 |
| | std | 1,97 | 2,70 | 0,88 | 0,53 |
| | se | 1,14 | 1,56 | 0,51 | 0,31 |
| 120 hours | average | 5,19 | 5,63 | 0,15 | 0,02 |
| | std | 0,49 | 0,88 | 0,08 | 0,24 |
| | se | 0,28 | 0,51 | 0,05 | 0,14 |
| 216 hours | average | 1,87 | 1,97 | 0,02 | -0,17 |
| | std | 0,62 | 0,79 | 0,17 | 0,36 |
| | se | 0,36 | 0,45 | 0,10 | 0,21 |
| 360 hours | average | 1,45 | 1,13 | 0,00 | -0,09 |
| | std | 0,63 | 0,80 | 0,13 | 0,23 |
| | se | 0,37 | 0,46 | 0,07 | 0,13 |
| 24 days | average | 0,88 | 0,79 | 0,10 | 0,11 |
| | std | 0,29 | 0,27 | 0,05 | 0,07 |
| | se | 0,17 | 0,16 | 0,03 | 0,04 |

| (b) | | µg N g-1 d-1 Mineralisation | µg N g-1 d-1 NH4 consumption | µg N g-1 d-1 Nitrification | µg N g-1 d-1 NO3 consumption |
|---|---|---|---|---|---|
| 24 hours | average | 32,45 | 29,98 | 1,11 | 0,84 |
| | std | 8,72 | 7,13 | 1,57 | 3,11 |
| | se | 5,04 | 4,12 | 0,91 | 1,79 |
| 72 hours | average | 12,27 | 12,97 | 3,21 | 3,41 |
| | std | 7,31 | 9,98 | 3,27 | 1,98 |
| | se | 4,22 | 5,76 | 1,89 | 1,14 |
| 120 hours | average | 19,23 | 20,84 | 0,55 | 0,07 |
| | std | 1,80 | 3,27 | 0,31 | 0,89 |
| | se | 1,04 | 1,89 | 0,18 | 0,51 |
| 216 hours | average | 6,94 | 7,31 | 0,08 | 0,00 |
| | std | 2,28 | 2,92 | 0,63 | 1,33 |
| | se | 1,32 | 1,68 | 0,36 | 0,77 |
| 360 hours | average | 5,36 | 4,18 | 0,00 | 0,00 |
| | std | 2,35 | 2,96 | 0,47 | 0,85 |
| | se | 1,36 | 1,71 | 0,27 | 0,49 |
| 24 days | average | 3,26 | 2,91 | 0,38 | 0,42 |
| | std | 1,06 | 1,00 | 0,19 | 0,26 |
| | se | 0,61 | 0,58 | 0,11 | 0,15 |

Values are mean of three plots (n = 3).

Table S2. Gross N process rates from vegetated surfaces (VP)

| (a) | | µg N cm-3 d-1
Mineralisation | µg N cm-3 d-1
NH4 consumption | µg N cm-3 d-1
Nitrification | µg N cm-3 d-1
NO3 conssumption |
|---|---|---|---|---|---|
| 24 hours | average | 0,00 | 0,00 | 0,02 | 0,04 |
| | std | 1,94 | 3,32 | 0,13 | 0,13 |
| | se | 1,12 | 1,92 | 0,08 | 0,08 |
| 72 hours | average | 0,55 | 0,43 | 0,00 | 0,01 |
| | std | 0,96 | 0,97 | 0,02 | 0,02 |
| | se | 0,55 | 0,56 | 0,01 | 0,01 |
| 120 hours | average | 1,15 | 1,11 | 0,00 | 0,00 |
| | std | 0,76 | 0,66 | 0,01 | 0,01 |
| | se | 0,44 | 0,38 | 0,01 | 0,01 |
| 216 hours | average | 0,63 | 0,63 | 0,00 | 0,00 |
| | std | 0,46 | 0,43 | 0,00 | 0,00 |
| | se | 0,26 | 0,25 | 0,00 | 0,00 |
| 360 hours | average | 0,37 | 0,36 | 0,00 | 0,00 |
| | std | 0,25 | 0,25 | 0,01 | 0,00 |
| | se | 0,15 | 0,14 | 0,00 | 0,00 |
| 24 days | average | 0,22 | 0,23 | 0,00 | 0,00 |
| | std | 0,18 | 0,15 | 0,00 | 0,00 |
| | se | 0,10 | 0,09 | 0,00 | 0,00 |

| (b) | | µg N g-1 d-1
Mineralisation | µg N g-1 d-1
NH4 consumption | µg N g-1 d-1
Nitrification | µg N g-1 d-1
NO3 consumption |
|---|---|---|---|---|---|
| 24 hours | average | 0,00 | 0,00 | 0,43 | 0,87 |
| | std | 38,70 | 66,35 | 2,61 | 2,61 |
| | se | 22,34 | 38,31 | 1,51 | 1,51 |
| 72 hours | average | 10,96 | 8,60 | 0,00 | 0,20 |
| | std | 19,21 | 19,42 | 0,45 | 0,45 |
| | se | 11,09 | 11,21 | 0,26 | 0,26 |
| 120 hours | average | 22,97 | 22,22 | 0,00 | 0,00 |
| | std | 15,15 | 13,15 | 0,18 | 0,18 |
| | se | 8,75 | 7,60 | 0,10 | 0,10 |
| 216 hours | average | 12,63 | 12,63 | 0,08 | 0,10 |
| | std | 9,18 | 8,62 | 0,08 | 0,08 |
| | se | 5,30 | 4,98 | 0,05 | 0,05 |
| 360 hours | average | 7,33 | 7,21 | 0,06 | 0,06 |
| | std | 5,10 | 4,93 | 0,14 | 0,05 |
| | se | 2,94 | 2,84 | 0,08 | 0,03 |
| 24 days | average | 4,46 | 4,62 | 0,07 | 0,08 |
| | std | 3,53 | 2,96 | 0,06 | 0,06 |
| | se | 2,04 | 1,71 | 0,03 | 0,03 |

Values are mean of three plots (n = 3).

---

## Author Response (AR1)

We thank the reviewers for their interest in our work and for their insightful comments that have greatly contributed to improve our manuscript. We have addressed the general and specific comments provided by the reviewers and have made necessary changes accordingly to their indications as follows:

**Comment on bg-2021-228**

**Anonymous Referee #1**

Referee comment on "Sources of nitrous oxide and fate of mineral nitrogen in sub-Arctic

permafrost peat soils" by Jenie A. Gil et al., Biogeosciences Discuss.,

https://doi.org/10.5194/bg-2021-228-RC1, 2021

The manuscript by Gil et al. addresses a pronounced research gap by investigating gross N turnover and inorganic N fates as well as N2O emissions in permafrost peatlands, including N2O source partitioning. They report much higher gross ammonification and nitrification for vegetation free compared to vegetated peatlands, which is explained e.g., by absent plant competition for N. Such detailed N cycle process knowledge is very scarce for permafrost ecosystems, which still makes prediction of permafrost nitrogen climate feedbacks highly uncertain. So this clearly is a timely study even if the experiments were already conducted more than 10 years ago. The overall manuscript quality is fine.

The quality of such field 15N studies to assess gross N turnover is strongly depending on a thorough experimental setup and this difficult task mostly appears to have been done very competent and thoroughly. On the other hand,

**(1) The chosen experimental setup with mirrored 15N labelling and all of its advantages and disadvantages was obviously designed to run the Ntrace model to estimate N turnover rates which then was not done. So the reader wonders, why not?**

A: The experimental design was chosen with the aim to quantify the relative contribution of nitrification and denitrification to the overall N2O fluxes from the bare peat surfaces (BP) using single and double 15N- labeled ammonium nitrate method, previously introduced by Baggs et al., (2003). Further, this experimental set-up allowed us to apply the traditional pool dilution technique to assess gross N transformation rates (mineralization, nitrification) in the field (Kirkham and Bartholomew, 1954). We considered that even though 15N tracing studies in combination with analyses via process-based models are the current "state-of-the-art" technique to quantify gross nitrogen (N) transformation rates and N2O emission pathways in soils, it would be very ambitious to apply the Ntrace model in an arctic peat soil in a field experiment as our first attempt to study N cycling in this soil. We wanted to first use a simpler approach and learn how the system behaves, before using the Ntrace model.

Although the Ntrace model has been used successfully for determination of gross nitrogen transformations in field studies and from organic soils (e.g., Holz *et al.*, 2015), it has never been successfully applied to study nitrogen cycling in situ in natural peatlands so far. This could be related to the fact that the accurate quantification of 15N-lable in the mineral N pools in organic soils is quite challenging (Mortland and Wolcott, 1965; Nõmmik and Vahtras,1982; Nieder et al., 2011). Additionally, high rate of microbial immobilization commonly causes problems for 15N tracing studies in soils with low mineral N availability, typical for arctic and sub-arctic soils: the 15N labeled mineral N can be very quickly immobilized and, thus, cannot be seen in the soil

extracts (Nordin, Schmidt & Shaver, 2004; Sørensen et al., 2008, Marushchak et al., 2021). In our study, the 15N label was recovered mostly in the bulk soil (up to 79%) during the entire experiment period, supporting the suggestion that biotic and/or abiotic process contribute to remove the 15N added from the extractable N pool. The uncertainties associated with these immobilization processes cause problems for the Ntrace model, and we need more knowledge about the fate of 15N after application before the model can be applied for peatlands in general, and Arctic peatlands in particular.

**(2) Due to addition of ammoniumnitrate in all treatments, gross nitrification rates are likely stimulated by substrate addition, which needs to be considered and discussed.**

A: The application rate of the label solutions was based on soil inorganic N concentrations measured from soils extracts from previous years (2007-2008). We have applied approx. 50% of the soil average inorganic N concentrations measured previously to avoid or minimize the effect of substrate adding on the soil N processes. The total amount also never exceeded maximum amounts measured in the soils. All this information is mentioned already in the manuscript (see below). Importantly, the contribution of N2O producing processes is likely not impacted, which was the main focus of the MS. We discussion this in the MS in page 20 lines 22-23 and page 21 lines 1 to 11:

"The concentration of inorganic N was at most doubled by adding labelled  $NO_3^-$  and/or  $NH_4^+$ , but the final nutrient content never exceeded maximum content of native  $NO_3^-$  or  $NH_4^+$  observed in the soil (data not shown). The bulk  $N_2O$ fluxes from the labelled plots (~10 mg  $N_2O$  m-2 d-1) were still within the range of  $N_2O$  fluxes observed in previous years from BP surfaces (1.9–31 mg  $N_2O$  m-2 d-1) (Repo *et al.*, 2009; Marushchak *et al.*, 2011). The differences in the  $N_2O$ fluxes from BP labelled and non-labelled plots could be also attributed to the natural spatial variation in the  $N_2O$  fluxes within the BP surfaces, which can be large even on small spatial scales (< 1m, personal observation. data not shown). The  $N_2O$  emissions from labelled and non-labelled plots had similar responses to changes in temperature (R2 = 0.391, p < 0.005), which was likely the major factor controlling the temporal variation in the  $N_2O$  fluxes from BP surfaces during the study period. Even if some stimulation occurred, this likely did not affect the relative contribution of different microbial pathways to the total  $N_2O$  emissions because BP surfaces were not N limited during the study period (see discussion below)."

**(3) Another issue could be that gross rates of N turnover were calculated based on day 1 to day 3 data with day 3 being a clear outlier in 15N recovery for bare peat (much lower than at day 1 and day 5; Fig. S1) – did this low 15N recovery lead to a bias in gross N turnover estimates, eventually because 15N was quickly leached in some labelling plots?**

A: The low recovery of 15N on day three was indeed surprising and a bit difficult to explain. We had teams which were responsible for labelling and sampling on different days, which could have potentially introduced some experimental artefacts and biased the results on total 15N recovery on day 3. However, the extractable N pools which were used to calculate gross N turnover rates and contributions of different processes to N2O emissions accounted only for a minor proportion of the total 15N pools and followed different dynamics (as in detail explained below). Thus, our main results were not biased by the low recovery of 15N on day 3, and we have sufficient confidence in the gross turnover estimates to report them here (more details below and in the following answer). It has to be also noted that if high leaching losses would have occurred in some labelling plots, mainly the calculations of N consumption would have been affected, and less so the calculations of production; the latter being one of the key aims of the study.

We added a sentence to the manuscript on page 15, line 1 to 3:

"At day 3, total recovery of 15N was lower than expected and although we have no explanation for these findings, this low recovery did not impact on the main results which were calculated from 15N in mineral nutrient pools."

To corroborate the reliability of the gross N turnover rates, we plot here a couple of more figures for the reviewer. In the figure S3a – b (in supplement file with answer to the referee), the natural logarithm of 15N atom percent excess of NH4+ is plotted against sampling time. Given constant rates this plot provides a linear relationship, while declines or increases in isotope pool dilution rates cause curvilinearity. For BP (blue cirlces) we can observe that curvilinearity starts after ~ day 9, this means that the transformation process rates are constant between 1 h and 9 days (R2 = 0.9125), so we can calculate and report values between this time period for BP. Gross transformation rates for VP were low (open squares) but there was also some linearity between day 1 and day 5 (R2 = 0.4195).

Results are more variable for 15N at% excess of NO3-, but there was a linear relationship over 24 to 72 hours (R2 =0,6957) for BP (Figure S3c - d). For VP surfaces, gross nitrification rates are negligible. High uncertainty is quite common in field labeling studies (e.g. Cookson et al., 2002; Harty et al., 2017). The high variability in our data could simply reflect the spatial variation at the site. Since constant process rates are a prerequisite for estimating gross N transformation rates (Kirkham and Bartholomew, 1954), we chose to report gross mineralization and nitrification rates for the period between 24 and 72 hours because gross mineralization and nitrification rates for BP were constant during this period (Figure S3). In addition, the changes in 15N at% excess of NO3-from day 5 (120 hours) in BP surfaces, suggest quick cycles of abiotic fixation and release of NO3-, as mention by the reviewer. To follow the recommendations from Braun et al., 2018 (and references therein) about shorter time period during pool dilution experiments to minimize errors due to remineralization of the label added, we decided not to include data from day 5 and beyond in the calculations of gross N process. We also observed low recovery of the 15N in the inorganic N pools at time 0 (1 hour after labeling) so we assumed that the 15N label was not well homogenized after 1h and did not use this time point for the calculations either.

Table S1 and S2 (in supplement file with answer to the referee) summarized gross mineralization and nitrification rates for all sampling points calculated from 24h for both surface types. We would like to mention that while re-calculating the data we have detected an error in our previous calculations of the gross N transformation rates and have now updated the rates. The gross rates have changed to some extent, but not largely. Our apologies for the previous mistake.

Looking at the table S1, the range of gross mineralization rate for BP at 72 hours is between 1,34 and 5.28  $\mu$ g N cm-3 d-1. This range included the mean rates calculated for 120 (5 days), 216 (9 days) and 360 hours (15 days). The same is observed for gross nitrification rates from BP (range from -0.01 to 1.75  $\mu$ g N cm-3 d-1), and gross mineralization rates from VP (Table S2, -0.41 to 1.51). Gross nitrification rates from VP are negligible. Consequently, we are confident that gross N transformation rates calculated for the period from 24 to 72 hours are representative of the gross N transformation rates for most of the experiment period. Exception to this are the gross rates calculated for 24 h and 24 days. To calculate the 24 hours rates, we used the data between 1h and 24h period. We have mention already that 15N label added was probably not well homogenized after 1h and we did not include this time point in the final calculations. For 24 days, probably too long period after adding the 15N label and some re-mineralization of the 15N label occurred resulting in smaller rates compared to previous days.

In the supplementary material in MS we have added two summary tables (Table S1 and Table S2) with gross N process for all sampling points for both surfaces and in both units  $\mu$ g N per g dry weight and  $\mu$ g cm-3 d-1. In the main text of the MS, we report and discuss gross N transformation rates calculated for the period between 24 to 72 hours. We have added the following explanation in in methods section 2.3.3 P 11 L 20- 28:

"The gross N transformation rates were calculated from data from T1 (15N-NO3-; nitrification) and T2 (15N-NH4+; ammonification) between time points 24 and 72 hours (3 d) after labeling. This time-period was selected because (1) gross nitrification rates for BP were constant during this period (Figure S3) and constant process rates are a prerequisite for estimating gross N transformation rates by Kirkham and Bartholomew, 1954 (2) the changes in 15N at % excess of NO3- from day 5 (120 hours) in BP surfaces, suggest quick cycles of abiotic fixation and release of NO3- (Figure 3 and S4), therefore shorter time period for the calculations is recommended to minimize errors due recycling of the label by assimilation to microbial biomass and remineralization (Braun et al., 2018) (3) the first time point of measurement (between 1 hour and 1 day after label application) could not be included in the calculations since that resulted often in negative gross N transformation, most likely because the label was not yet evenly distributed in the soil"

(4) Further, the temporal dynamics of 15N recovery in the nitrate pool after 15N-nitrate labelling is problematic. Data show that there is an increase in recovery between day 0 and day1, a decrease between day 1 and day 3 which is used to calculate gross ammonification, followed by another increase. I suppose therefore that atom%15N enrichment of nitrate also shows no persistent dilution. Hence, choosing other time steps for calculating gross nitrification might reveal completely different results or even negative rates. Were there probably quick cycles of abiotic fixation and release of nitrate? Or is this originating from problems with 15N labelling as described above? Based on these thoughts it appears to me that gross nitrification rates in this study might be pretty unreliable. Considering this would require major changes in the discussion section.

A: We have mostly answered to these questions of the reviewer in the previous comment. In a nutshell: due to drawbacks associated to the study of organic rich soils, such as peat soils, under field conditions using pool dilution technique, the calculations of the gross nitrogen transformation rates were indeed not straightforward. We base our decision to report gross nitrification rates between day 1 and 3 on the fact that the label is not homogeneous distributed before day 1 and that there is re-cycling of labelling after day 3. Importantly, there is a linear relationship between ln of 15N in nutrients and time, thus the analysis gives reliable results. We have backed up these assumptions with more figures and tables in the supplementary, and explanations in the text as well as relevant literature (previous comment, main MS). We acknowledge that the uncertainty is large, particularly for gross nitrification rates. However, since very few data have been published on gross nitrification rates from arctic and sub-arctic soils, we considered that our results are still valuable to the scientific community.

We further note that the results we present on gross mineralization and nitrification rates fall within the range of published gross turnover rates for Arctic soils (though only a few exist for gross nitrification) (e.g. Wild *et al.*, 2015). Also, all data were sampled and analyzed the same way, thus even if there was some variability within the data and the choice of time points for the gross turnover calculations is associated with some assumptions, the comparison between VP and BP which was the main focus here, is still valid.

**P3 L 6:** The sentence "Denitrification releases usually more N2O under wetter, more anaerobic conditions..." should be further specified as it is otherwise misleading. Under very anaerobic conditions N2O emissions by denitrification are expected to decline as denitrification until the terminal product N2 is favored.

A: thanks, we modified the text accordingly, and the new sentence reads now:

"Compared to nitrification, denitrification releases usually more N2O under wetter, more anaerobic conditions and has been suggested as the key process for N2O production in bare peat surfaces (Repo *et al.*, 2009)."

**P 3 L 23:** Do you mean microbial immobilization? Please specify.

A: Yes, modified accordingly in the MS text.

**P 5 L 30** probably a few more details how gas samples were transferred (overpressure? Preevacuated vials?)

A: Ok, few more details were added:

"Gas samples (20 mL) were transferred into 12 ml pre-evacuated exetainers equipped with butyl rubber septa (Labco Ltd, UK) the same day of sampling."

**P 5 L32** Leakage test with a standard gas can be conducted only for other vials.

A: Yes, we cannot test the exact vials with samples. However, we took around 10-15 random vials from the same batch we were going to use for sampling and used the same protocol for sample collection (e.g same lids, evacuation time, sample volume) but filled those with N2O standard. We carried those to the field with us and brought them back for analysis together with field samples.

**P5 L35**: the authors write that N2O emissions calculated based on only two concentration measurements were compared with adjacent static chambers that had higher sampling frequency. Good, but what was the outcome of this comparison?

A: Also reviewer 2 mentioned that this point was not clear, and we are sorry for that. We thus repeat the answer to reviewer 2 here for reviewer 1: The comparison between small chamber, non-permanent, one-point sampling (labeled plots) and the large chambers on permanent plots with multiple samplings points (non-labeled plots) is shown graphically in figure 1 in the MS. We are sorry that it was not clear and have changed the caption and description of figure 1. N2O fluxes from labeled plots were higher compared to N2O fluxes from non-labelled plots but never higher that the highest N2O fluxes measured on previous years.

In addition, for clarification here, we have calculated N2O fluxes from three random large chambers on permanent plots by taking only the last sampling point (60 min) and compared those to N2O fluxes calculated by linear integration over all sampling points (5, 20, 45 and 60 min) (Table A). We observed that the difference between the two approaches was between 1 to 5 %. Even with  $\pm$  5% overestimation, N2O fluxes from labeled plots (small, non-permanent chambers) are in the range of N2O fluxes from non-labeled plots from previous years.

|           | 1 01     | •                  | 1 01                  |                |             |
|-----------|----------|--------------------|-----------------------|----------------|-------------|
|           |          |                    | linear interplolation | end-time point | % diference |
| Chamber 1 | N2O Flux | (µg N2O-N m-2 h-1) | 19,0                  | 19,9           | 5           |
| Chaber 2  | N2O Flux | (µg N2O-N m-2 h-1) | 10,6                  | 10,7           | 0           |
| Chaber 3  | N2O Flux | (µg N2O-N m-2 h-1) | 58,9                  | 58,6           | 1           |

**Table A.** N2O fluxes from larger chambers on non-labeled plots calculated by linear integration over all sampling points and only the last sampling point.

Usually pushing the chamber into the soil disturbs mainly the root system, but there were not plants and roots in BP surfaces roots. There are several papers presenting N2O fluxes from non-permanent chambers, by pushing the chamber into the soil (e.g Weitz *et al.*, 1999; Maljanen *et al.*, 2007; Hyvönen *et al.*, 2009).

**P8 L18** The abbreviation "T" for treatment might be misunderstood by the reader as extraction time steps, so probably choose another abbreviation for treatment.

A: We hope that we explain the abbreviations sufficiently well in the MS. We would like to keep thus the original version right now, but if the reviewer and editor insist, we are ready to change it in another review round.

**Statistics section**: Only N2O mentioned, what about gross N turnover? **A:** It is now added in the text.

**P9 L 18:** N2O measurements are reported for a little more than a month – I therefore would not speak of "seasonal patterns"

A: agreed, we deleted the word seasonal from the sentences.

**Figure 3**: It shows 15N excess? The caption states that it is 15N enrichment, which is not true. Please clarify. Generally, the 15NO3- seems quite problematic and cannot be explained by gross nitrification but probably by 15N nitrate fixation and release in organic matter? This would question the gross nitrification calculations – showing the atom% excess enrichment would further help to judge this. A thorough and critical discussion is needed here.

A: The reviewer is correct. The caption of the figure 3 has been corrected. It is easier to follow the changes in the movement of the 15N excess from the figure S4 (in supplement file with answer to the referee). The 15N at% excess values of NO3- and NH4+ behaved in general as expected from the pool dilution experiment. The changes in 15N at% excess of NO3- from day 5 (120 hours) in BP surfaces, suggest quick cycles of abiotic fixation and release of NO3-, as mention by the reviewer. Consequently, we chose to report gross mineralization and nitrification rates for the period between 24 and 72 hours as explained in previous comments.

In the discussion section of the MS (section 4.3, P22 L 10–30, P23 L 1–6) we have acknowledged that biotic and/or abiotic process could contribute to remove the 15N added from the extractable N pool and that the uncertainties associated with these immobilization processes could have an impact particularly on the gross nitrification rates calculations.

We have included in supplementary material Figure S4 and Table S1 and S2 showing the gross N transformation rates for the whole study period.

**P13 L 18**: "...27 and 90 times higher.." – sure, dividing by very small rates is giving such impressive numbers which are however a bit misleading as rates at VP were hardly present. And: please give the rates related to soil dry weight as well not only related to cm3.

A: The reviewer is correct, particularly gross nitrification rates from VP are negligible.

NO3- consumption is not zero (see table S1 and S2). It is small but no zero. We did set to zero all negative values.

We have modified the text in MS as follow in results section 3.5 (P19 L10-21):

"As described above in the methods section 2.3.3, we chose to report gross mineralization and nitrification rates for the period between 24 and 72 hours. During this time period, the gross mineralization and nitrification rates were constant and positive, and we could assume negligible recycling of the 15N label via microbial biomass back to the mineral N pool (Braun et al., 2018). For method comparison purposes, we have shown the gross mineralization and nitrification rates calculated for different periods in table S1 and S2 in supplementary material. We note that variability in the results depending on the time period chosen for the calculations was higher for gross nitrification rates than for gross mineralization rendering higher uncertainties in the nitrification data. However, the comparison between VP and BP, which is the focus here, is independent of the chosen calculation method and is valid. Further, high variability of gross N turnover rates is quite common in field labeling studies (e.g., Cookson et al., 2002; Harty et al., 2017). The high

variability in our data could also simply reflect the spatial variation at the site between the subplots destructively sampled at different time-points.

Gross mineralization and nitrification rates in BP were higher than in VP (p < 0.01) (Table 2). In BP, gross mineralization rates were four times higher than gross nitrification rates. Gross nitrification rates in VP surfaces were negligible. NH4+ consumption rates were similar to gross mineralization rates for both surface types and higher in BP, while NO3- consumption only took place in BP surface and not in VP. See Table S1 and S2 in supplementary material, for gross N transformation rates calculated on a soil weight basis."

**References**

Baggs, E. M., Richter, M., Cadisch, G., and Hartwig, U. A.: Denitrification in grass swards is increased under elevated atmospheric CO2, Soil Biology & Biochemistry, 35, 729-732, 10.1016/s0038-0717(03)00083-x, 2003.

Braun, J., Mooshammer, M., Wanek, W., Prommer, J., Walker, T. W. N., Rutting, T., and Richter, A.: Full N-15 tracer accounting to revisit major assumptions of N-15 isotope pool dilution approaches for gross nitrogen mineralization, Soil Biology & Biochemistry, 117, 16-26, 10.1016/j.soilbio.2017.11.005, 2018.

Cookson, W., Cornforth, I.S, Rowarth, J.S,: Winter soil temperature (2–15°C) effects on nitrogen transformations in clover green manure amended or unamended soils; a laboratory and field study, Soil Biology and Biochemistry, Volume 34, Issue 10, Pages 1401-1415, ISSN 0038-0717, https://doi.org/10.1016/S0038-0717(02)00083-4, 2002.

Harty, M.A., McGeough, K.L., Carolan, R., Müller, C., Laughlin, R.J., Lanigan, G.J., Richards, K.G., Watson, C.J. : Gross nitrogen transformations in grassland soil react differently to urea stabilisers under laboratory and field conditions, Soil Biology and Biochemistry, Volume 109, Pages 23-34, ISSN 0038-0717, https://doi.org/10.1016/j.soilbio.2017.01.025, 2017.

Holz, M., Aurangojeb, M., Kasimir, Å. et al. Gross Nitrogen Dynamics in the Mycorrhizosphere of an Organic Forest Soil. Ecosystems 19, 284–295. https://doi.org/10.1007/s10021-015-9931-4, 2016.

Hyvönen, N.P., Huttunen, J.T., Shurpali, N.J., Tavi, N.M., Repo, M.E., Martikainen, P.J. : Fluxes of nitrous oxide and methane on an abandoned peat extraction site: Effect of reed canary grass cultivation, Bioresource Technology, Volume 100, Issue 20, Pages 4723-4730, ISSN 0960-8524, https://doi.org/10.1016/j.biortech.2009.04.043, 2009.

Kirkham, D. and Bartholomew, W.: Equations for following nutrient transformations in soil, utilizing tracer data, Soil Science Society of America Journal, 18, 33-34, 1954.

Maljanen, M., Martikkala, M., Koponen, H.T., Virkajärvi, P. and Martikainen, P.J: Fluxes of nitrous oxide and nitric oxide from experimental excreta patches in boreal agricultural soil, Soil Biology and Biochemistry, Volume 39, Issue 4, Pages 914-920, ISSN 0038-0717, https://doi.org/10.1016/j.soilbio.2006.11.001, 2007.

Marushchak, M.E., Kerttula, J., Diáková, K. et al.: Thawing Yedoma permafrost is a neglected nitrous oxide source. Nat Commun 12, 7107. https://doi.org/10.1038/s41467-021-27386-2, 2021

Mortland, M.M. and Wolcott, A.R. Sorption of Inorganic Nitrogen Compounds by Soil Materials. In Soil Nitrogen (eds W.V. Bartholomew and F.E. Clark). https://doi.org/10.2134/agronmonogr10.c4, 1965. Nieder, R., Benbi, D.K. & Scherer, H.W. Fixation and defixation of ammonium in soils: a review. Biol Fertil Soils 47, 1–14. https://doi.org/10.1007/s00374-010-0506-4, 2011.

Nommik, H. and Vahtras, K. Retention and Fixation of Ammonium and Ammonia in Soils. In Nitrogen in Agricultural Soils, F.J. Stevenson (Ed.). https://doi.org/10.2134/agronmonogr22.c4, 1982.

Nordin, A., Schmidt, I. K., and Shaver, G. R.: Nitrogen uptake by arctic soil microbes and plants in relation to soil nitrogen supply, Ecology, 85, 955-962, 10.1890/03-0084, 2004.

Sorensen, P. L., Clemmensen, K. E., Michelsen, A., Jonasson, S., and Strom, L.: Plant and microbial uptake and allocation of organic and inorganic nitrogen related to plant growth forms and soil conditions at two subarctic tundra sites in Sweden, Arctic Antarctic and Alpine Research, 40, 171-180, 10.1657/1523-0430(06-114;2, 2008.

Weitz, A. M., Keller, M., Linder, E., and Crill, P. M., Spatial and temporal variability of nitrogen oxide and methane fluxes from a fertilized tree plantation in Costa Rica, J. Geophys. Res., 104 (D23), 30097–30107, doi:10.1029/1999JD900952, 1999.

Wild, B., Schnecker, J., Knoltsch, A., Takriti, M., Mooshammer, M., Gentsch, N., Mikutta, R., Alves, R. J. E., Gittel, A., Lashchinskiy, N., and Richter, A.: Microbial nitrogen dynamics in organic and mineral soil horizons along a latitudinal transect in western Siberia, Global Biogeochemical Cycles, 29, 567-582, 10.1002/2015gb005084, 2015.

**Comment on bg-2021-228**

**Anonymous Referee #1 – Figures and tables**

**Figure S3**. Change in natural logarithmic of 15N at% of (a - b) NH4+ an (c - d) NO3- in BP and VP soil over total study period.